# Embedding Space Interpolation Beyond Mini-Batch, Beyond Pairs and Beyond Examples

Shashanka Venkataramanan[1]    Ewa Kijak[1]    Laurent Amsaleg[1]    Yannis Avrithis[2]

[1]Inria, Univ Rennes, CNRS, IRISA
[2]Institute of Advanced Research on Artificial Intelligence (IARAI)

## Abstract

*Mixup* refers to interpolation-based data augmentation, originally motivated as a way to go beyond *empirical risk minimization* (ERM). Its extensions mostly focus on the definition of interpolation and the space (input or embedding) where it takes place, while the augmentation process itself is less studied. In most methods, the number of generated examples is limited to the mini-batch size and the number of examples being interpolated is limited to two (pairs), in the input space.

We make progress in this direction by introducing *MultiMix*, which generates an arbitrarily large number of interpolated examples beyond the mini-batch size, and interpolates the entire mini-batch in the embedding space. Effectively, we sample on the entire *convex hull* of the mini-batch rather than along linear segments between pairs of examples.

On sequence data we further extend to *Dense MultiMix*. We densely interpolate features and target labels at each spatial location and also apply the loss densely. To mitigate the lack of dense labels, we inherit labels from examples and weight interpolation factors by attention as a measure of confidence.

Overall, we increase the number of loss terms per mini-batch by orders of magnitude at little additional cost. This is only possible because of interpolating in the *embedding space*. We empirically show that our solutions yield significant improvement over state-of-the-art mixup methods on four different benchmarks, despite interpolation being only linear. By analyzing the embedding space, we show that the classes are more tightly clustered and uniformly spread over the embedding space, thereby explaining the improved behavior.

## 1   Introduction

*Mixup* [65] is a data augmentation method that interpolates between pairs of training examples, thus regularizing a neural network to favor linear behavior in-between examples. Besides improving generalization, it has important properties such as reducing overconfident predictions and increasing the robustness to adversarial examples. Several follow-up works have studied interpolation in the *latent* or *embedding* space, which is equivalent to interpolating along a manifold in the input space [51], and a number of nonlinear and attention-based interpolation mechanisms [62, 26, 25, 46, 7]. However, little progress has been made in the augmentation process itself, *i.e.*, the number $n$ of generated examples and the number $m$ of examples being interpolated.

Mixup was originally motivated as a way to go beyond *empirical risk minimization* (ERM) [47] through a vicinal distribution expressed as an expectation over an interpolation factor $\lambda$, which is equivalent to the set of linear segments between all pairs of training inputs and targets. In practice however, in every training iteration, a single scalar $\lambda$ is drawn and the number of interpolated pairs is limited to the size $b$ of the mini-batch ($n = b$), as illustrated in Figure 1(a). This is because if

37th Conference on Neural Information Processing Systems (NeurIPS 2023).

| METHOD | SPACE | TERMS | MIXED | FACT | DISTR |
|---|---|---|---|---|---|
| Mixup [65] | input | $b$ | 2 | 1 | Beta |
| Manifold mixup [51] | embedding | $b$ | 2 | 1 | Beta |
| $\zeta$-Mixup [1] | input | $b$ | 25 | 1 | RandPerm |
| SuperMix [11] | input | $b$ | 3 | 1 | Dirichlet |
| MultiMix (ours) | embedding | $n$ | $b$ | $n$ | Dirichlet |
| Dense MultiMix (ours) | embedding | $nr$ | $b$ | $nr$ | Dirichlet |

Table 1: *Interpolation method properties*. SPACE: Space where interpolation takes place; TERMS: number of loss terms per mini-batch; MIXED: maximum number $m$ of examples being interpolated; FACT: number of interpolation factors $\lambda$ per mini-batch; DISTR: distribution used to sample interpolation factors; RandPerm: random permutations of a fixed discrete probability distribution. $b$: mini-batch size; $n$: number of generated examples per mini-batch; $r$: spatial resolution.

interpolation takes place in the input space, it would be expensive to increase the number of pairs per iteration. To our knowledge, these limitations exist in all mixup methods.

In this work, we argue that a data augmentation process should increase the data seen by the model, or at least by its last few layers, as much as possible. In this sense, we follow *manifold mixup* [51] and generalize it in a number of ways to introduce *MultiMix*.

First, we increase the number $n$ of generated examples beyond the mini-batch size $b$, by orders of magnitude ($n \gg b$). This is possible by interpolating at the deepest layer, *i.e.*, just before the classifier, which happens to be the most effective choice. To our knowledge, we are the first to investigate $n > b$.

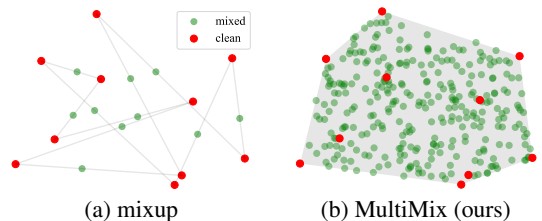

(a) mixup          (b) MultiMix (ours)

Figure 1: Data augmentation in a mini-batch $B$ of $b = 10$ points in two dimensions. (a) *mixup*: sampling $n = b$ points on linear segments between $b$ pairs of points using the same interpolation factor $\lambda$. (b) *MultiMix*: sampling $n = 300$ points in the convex hull of $B$.

Second, we increase the number $m$ of examples being interpolated from $m = 2$ (pairs) to $m = b$ (a single *tuple* containing the entire mini-batch). Effectively, instead of linear segments between pairs of examples in the mini-batch, we sample on their entire *convex hull* as illustrated in Figure 1(b). This idea has been investigated in the input space: the original mixup method [65] found it non-effective, while [11] found it effective only up to $m = 3$ examples and [1] went up to $m = 25$ but with very sparse interpolation factors. To our knowledge, we are the first to investigate $m > 2$ in the embedding space and to show that it is effective up to $m = b$.

Third, instead of using a single scalar value of $\lambda$ per mini-batch, we draw a different vector $\lambda \in \mathbb{R}^m$ for each interpolated example. A single $\lambda$ works for standard mixup because the main source of randomness is the choice of pairs (or small tuples) out of $b$ examples. In our case, because we use a single tuple of size $m = b$, the only source of randomness being $\lambda$.

We also argue that, what matters more than the number of (interpolated) examples is the total number of *loss terms* per mini-batch. A common way to increase the number of loss terms per example is by *dense* operations when working on sequence data, *e.g.* patches in images or voxels in video. This is common in dense tasks like segmentation [41] and less common in classification [27]. We are the first to investigate this idea in mixup, introducing *Dense MultiMix*.

In particular, this is an extension of MultiMix where we work with feature tensors of spatial resolution $r$ and densely interpolate features and targets at each spatial location, generating $r$ interpolated features per example and $nr > n$ per mini-batch. We also apply the loss densely. This increases the number of loss terms further by a factor $r$, typically one or two orders of magnitude, compared with MultiMix. Of course, for this to work, we also need a target label per feature, which we inherit from the corresponding example. This is a *weak* form of supervision [70]. To carefully select the most representative features per object, we use an *attention map* representing our confidence in the target label per spatial location. The interpolation vectors $\lambda$ are then weighted by attention.

Table 1 summarizes the properties of our solutions against existing interpolation methods. Overall, we make the following contributions:

1. We generate an *arbitrary large number of interpolated examples* beyond the mini-batch size, each by interpolating the entire mini-batch in the embedding space, with one interpolation vector per example. (subsection 3.2).

2. We extend to attention-weighted *dense* interpolation in the embedding space, further increasing the number of loss terms per example (subsection 3.3).

3. We improve over state-of-the-art (SoTA) mixup methods on *image classification*, *robustness to adversarial attacks*, *object detection* and *out-of-distribution detection*. Our solutions have little or no additional cost while interpolation is only linear (section 4).

4. Analysis of the embedding space shows that our solutions yield classes that are *tightly clustered* and *uniformly spread* over the embedding space (section 4).

## 2   Related Work

**Mixup: interpolation methods**   In general, mixup interpolates between pairs of input examples [65] or embeddings [51] and their corresponding target labels. Several follow-up methods mix input images according to spatial position, either at random rectangles [62] or based on attention [46, 26, 25], in an attempt to focus on a different object in each image. Other follow-up method [35] randomly cuts image at patch level and obtains its corresponding mixed label using content-based attention. We also use attention in our dense MultiMix variant, but in the embedding space. Other definitions of interpolation include the combination of content and style from two images [23], generating out-of-manifold samples using adaptive masks [37], generating binary masks by thresholding random low-frequency images [18] and spatial alignment of dense features [48]. Our dense MultiMix also uses dense features but without aligning them and can interpolate a large number of samples, thereby increasing the number of loss-terms per example. Our work is orthogonal to these methods as we focus on the sampling process of augmentation rather than on the definition of interpolation. We refer the reader to [30] for a comprehensive study of mixup methods.

**Mixup: number of examples to interpolate**   To the best of our knowledge, the only methods that interpolate more than two examples for image classification are OptTransMix [71], SuperMix [11], $\zeta$-Mixup [1] and DAML [45]. All these methods operate in the *input space* and limit the number of generated examples to the mini-batch size; whereas MultiMix generates an arbitrary number of interpolated examples (more than $1000$ in practice) in the *embedding space*. To determine the interpolation weights, OptTransMix uses a complex optimization process and only applies to images with clean background; $\zeta$-Mixup uses random permutations of a fixed vector; SuperMix uses a Dirichlet distribution over *not more than* $3$ examples in practice; and DAML uses a Dirichlet distribution to interpolate across different domains. We also use a Dirichlet distribution but over as many examples as the mini-batch size. Similar to MultiMix, [6] uses different $\lambda$ within a single batch. Beyond classification, $m$-Mix [66] uses graph neural networks in a self-supervised setting with pair-based loss functions. The interpolation weights are deterministic and based on pairwise similarities.

**Number of loss terms**   Increasing the number of loss terms more than $b$ per mini-batch is not very common in classification. In *deep metric learning* [40, 28, 53], it is common to have a loss term for each pair of examples in a mini-batch in the embedding space, giving rise to $b(b-1)/2$ loss terms per mini-batch, even without mixup [49]. *Contrastive* loss functions are also common in self-supervised learning [8, 5] but have been used for supervised image classification too [24]. Ensemble methods [55, 14, 19, 3] increase the number of loss terms, but with the proportional increase of the cost by operating in the input space. To our knowledge, MultiMix is the first method to increase the number of loss terms per mini-batch to $n \gg b$ for mixup. Dense MultiMix further increases this number to $nr$. By operating in the embedding space, their computational overhead is minimal.

**Dense loss functions**   Although standard in dense tasks like semantic segmentation [41, 20], where dense targets commonly exist, dense loss functions are less common otherwise. Few examples are in *weakly-supervised segmentation* [70, 2], *few-shot learning* [33, 31], where data augmentation is of utter importance, *attribution methods* [27] and *unsupervised representation learning*, *e.g.* dense contrastive learning [42, 54], learning from spatial correspondences [59, 57] and masked language or image modeling [12, 58, 32, 69]. To our knowledge, we are the first to use dense interpolation and a dense loss function for mixup.

# 3 Method

## 3.1 Preliminaries and background

**Problem formulation** Let $x \in \mathcal{X}$ be an input example and $y \in \mathcal{Y}$ its one-hot encoded target, where $\mathcal{X} = \mathbb{R}^D$ is the input space, $\mathcal{Y} = \{0,1\}^c$ and $c$ is the total number of classes. Let $f_\theta : \mathcal{X} \to \mathbb{R}^d$ be an encoder that maps the input $x$ to an embedding $z = f_\theta(x)$, where $d$ is the dimension of the embedding. A classifier $g_W : \mathbb{R}^d \to \Delta^{c-1}$ maps $z$ to a vector $p = g_W(z)$ of predicted probabilities over classes, where $\Delta^n \subset \mathbb{R}^{n+1}$ is the unit $n$-simplex, *i.e.*, $p \geq 0$ and $\mathbf{1}_c^\top p = 1$, and $\mathbf{1}_c \in \mathbb{R}^c$ is an all-ones vector. The overall network mapping is $f := g_W \circ f_\theta$. Parameters $(\theta, W)$ are learned by optimizing over mini-batches.

Given a mini-batch of $b$ examples, let $X = (x_1, \ldots, x_b) \in \mathbb{R}^{D \times b}$ be the inputs, $Y = (y_1, \ldots, y_b) \in \mathbb{R}^{c \times b}$ the targets and $P = (p_1, \ldots, p_b) \in \mathbb{R}^{c \times b}$ the predicted probabilities of the mini-batch, where $P = f(X) := (f(x_1), \ldots, f(x_b))$. The objective is to minimize the cross-entropy

$$H(Y, P) := -\mathbf{1}_c^\top (Y \odot \log(P)) \mathbf{1}_b / b \tag{1}$$

of predicted probabilities $P$ relative to targets $Y$ averaged over the mini-batch, where $\odot$ is the Hadamard (element-wise) product. In summary, the mini-batch loss is

$$L(X, Y; \theta, W) := H(Y, g_W(f_\theta(X))). \tag{2}$$

The total number of loss terms per mini-batch is $b$.

**Mixup** Mixup methods commonly interpolate pairs of inputs or embeddings and the corresponding targets at the mini-batch level while training. Given a mini-batch of $b$ examples with inputs $X$ and targets $Y$, let $Z = (z_1, \ldots, z_b) \in \mathbb{R}^{d \times b}$ be the embeddings of the mini-batch, where $Z = f_\theta(X)$. *Manifold mixup* [51] interpolates the embeddings and targets by forming a convex combination of the pairs with interpolation factor $\lambda \in [0, 1]$:

$$\widetilde{Z} = Z(\lambda I + (1 - \lambda)\Pi) \tag{3}$$

$$\widetilde{Y} = Y(\lambda I + (1 - \lambda)\Pi), \tag{4}$$

where $\lambda \sim \mathrm{Beta}(\alpha, \alpha)$, $I$ is the identity matrix and $\Pi \in \mathbb{R}^{b \times b}$ is a permutation matrix. *Input mixup* [65] interpolates inputs rather than embeddings:

$$\widetilde{X} = X(\lambda I + (1 - \lambda)\Pi). \tag{5}$$

Whatever the interpolation method and the space where it is performed, the interpolated data, *e.g.* $\widetilde{X}$ [65] or $\widetilde{Z}$ [51], replaces the original mini-batch data and gives rise to predicted probabilities $\widetilde{P} = (p_1, \ldots, p_b) \in \mathbb{R}^{c \times b}$ over classes, *e.g.* $\widetilde{P} = f(\widetilde{X})$ [65] or $\widetilde{P} = g_W(\widetilde{Z})$ [51]. Then, the average cross-entropy $H(\widetilde{Y}, \widetilde{P})$ (1) between the predicted probabilities $\widetilde{P}$ and interpolated targets $\widetilde{Y}$ is minimized. The number of generated examples per mini-batch is $n = b$, same as the original mini-batch size, and each is obtained by interpolating $m = 2$ examples. The total number of loss terms per mini-batch is again $b$.

## 3.2 MultiMix

**Interpolation** The number of generated examples per mini-batch is now $n \gg b$ and the number of examples being interpolated is $m = b$. Given a mini-batch of $b$ examples with embeddings $Z$ and targets $Y$, we draw interpolation vectors $\lambda_k \sim \mathrm{Dir}(\alpha)$ for $k = 1, \ldots, n$, where $\mathrm{Dir}(\alpha)$ is the symmetric Dirichlet distribution and $\lambda_k \in \Delta^{m-1}$, that is, $\lambda_k \geq 0$ and $\mathbf{1}_m^\top \lambda_k = 1$. We then interpolate embeddings and targets by taking $n$ convex combinations over all $m$ examples:

$$\widetilde{Z} = Z\Lambda \tag{6}$$

$$\widetilde{Y} = Y\Lambda, \tag{7}$$

where $\Lambda = (\lambda_1, \ldots, \lambda_n) \in \mathbb{R}^{b \times n}$. We thus generalize manifold mixup [51]:

1. from $b$ to an arbitrary number $n \gg b$ of generated examples: interpolated embeddings $\widetilde{Z} \in \mathbb{R}^{d \times n}$ (6) *vs.* $\mathbb{R}^{d \times b}$ in (3), targets $\widetilde{Y} \in \mathbb{R}^{c \times n}$ (7) *vs.* $\mathbb{R}^{c \times b}$ in (4);

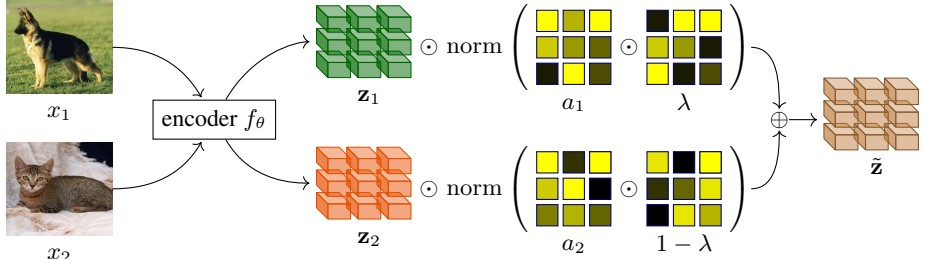

Figure 2: *Dense MultiMix* (subsection 3.3) for the special case $m = 2$ (two examples), $n = 1$ (one interpolated embedding), $r = 9$ (spatial resolution $3 \times 3$). The embeddings $\mathbf{z}_1, \mathbf{z}_2 \in \mathbb{R}^{d \times 9}$ of input images $x_1, x_2$ are extracted by encoder $f_\theta$. Attention maps $a_1, a_2 \in \mathbb{R}^9$ are extracted (9), multiplied element-wise with interpolation vectors $\lambda, (1 - \lambda) \in \mathbb{R}^9$ (10) and $\ell_1$-normalized per spatial position (11). The resulting weights are used to form the interpolated embedding $\tilde{\mathbf{z}} \in \mathbb{R}^{d \times 9}$ as a convex combination of $\mathbf{z}_1, \mathbf{z}_2$ per spatial position (12). Targets are interpolated similarly (13).

2. from pairs ($m = 2$) to a tuple of length $m = b$, containing the entire mini-batch: $m$-term convex combination (6),(7) *vs.* 2-term in (3),(4), Dirichlet *vs.* Beta distribution;

3. from fixed $\lambda$ across the mini-batch to a different $\lambda_k$ for each generated example.

**Loss**     Again, we replace the original mini-batch embeddings $Z$ by the interpolated embeddings $\widetilde{Z}$ and minimize the average cross-entropy $H(\widetilde{Y}, \widetilde{P})$ (1) between the predicted probabilities $\widetilde{P} = g_W(\widetilde{Z})$ and the interpolated targets $\widetilde{Y}$ (7). Compared with (2), the mini-batch loss becomes

$$L_M(X, Y; \theta, W) := H(Y\Lambda, g_W(f_\theta(X)\Lambda)). \tag{8}$$

The total number of loss terms per mini-batch is now $n \gg b$.

### 3.3   Dense MultiMix

We now extend to the case where the embeddings are structured, *e.g.* in tensors. This happens *e.g.* with token *vs.* sentence embeddings in NLP and patch *vs.* image embeddings in vision. It works by removing spatial pooling and applying the loss function densely over all tokens/patches. The idea is illustrated in Figure 2. For the sake of exposition, our formulation uses sets of matrices grouped either by example or by spatial position. In practice, all operations are on tensors.

**Preliminaries**     The encoder is now $f_\theta : \mathcal{X} \to \mathbb{R}^{d \times r}$, mapping the input $x$ to an embedding $\mathbf{z} = f_\theta(x) \in \mathbb{R}^{d \times r}$, where $d$ is the number of channels and $r$ is its spatial resolution—if there are more than one spatial dimensions, these are flattened.

Given a mini-batch of $b$ examples, we have again inputs $X = (x_1, \dots, x_b) \in \mathbb{R}^{D \times b}$ and targets $Y = (y_1, \dots, y_b) \in \mathbb{R}^{c \times b}$. Each embedding $\mathbf{z}_i = f_\theta(x_i) = (z_i^1, \dots, z_i^r) \in \mathbb{R}^{d \times r}$ for $i = 1, \dots, b$ consists of features $z_i^j \in \mathbb{R}^d$ for spatial position $j = 1, \dots, r$. We group features by position in matrices $Z^1, \dots, Z^r$, where $Z^j = (z_1^j, \dots, z_b^j) \in \mathbb{R}^{d \times b}$ for $j = 1, \dots, r$.

**Attention**     In the absence of dense targets, each spatial location inherits the target of the corresponding input example. This is weak supervision, because the target object is not visible everywhere. To select the most reliable locations, we define a level of confidence according to an attention map. Given an embedding $\mathbf{z} \in \mathbb{R}^{d \times r}$ with target $y \in \mathcal{Y}$ and a vector $u \in \mathbb{R}^d$, the *attention map*

$$a = h(\mathbf{z}^\top u) \in \mathbb{R}^r \tag{9}$$

measures the similarity of features of $\mathbf{z}$ to $u$, where $h$ is a non-linearity, *e.g.* softmax or ReLU followed by $\ell_1$ normalization. There are different ways to define vector $u$. For example, $u = \mathbf{z}\mathbf{1}_r/r$ by global average pooling (GAP) of $\mathbf{z}$, or $u = Wy$ assuming a linear classifier with $W \in \mathbb{R}^{d \times c}$, similar to class activation mapping (CAM) [68]. In case of no attention, $a = \mathbf{1}_r/r$ is uniform.

Given a mini-batch, let $a_i = (a_i^1, \dots, a_i^r) \in \mathbb{R}^r$ be the attention map of embedding $\mathbf{z}_i$ (9) for $i = 1, \dots, b$. We group attention by position in vectors $a^1, \dots, a^r$, where $a^j = (a_1^j, \dots, a_b^j) \in \mathbb{R}^b$

for $j = 1, \ldots, r$. Figure 6 in the supplementary shows the attention obtained by (9). We observe high confidence on the entire or part of the object. Where confidence is low, we assume the object is not visible and thus the corresponding interpolation factor should be low.

**Interpolation** There are again $n \gg b$ generated examples per mini-batch, with $m = b$ examples being densely interpolated. For each spatial position $j = 1, \ldots, r$, we draw interpolation vectors $\lambda_k^j \sim \mathrm{Dir}(\alpha)$ for $k = 1, \ldots, n$ and define $\Lambda^j = (\lambda_1^j, \ldots, \lambda_n^j) \in \mathbb{R}^{m \times n}$. Since input examples are assumed to contribute according to the attention vector $a^j \in \mathbb{R}^m$, we scale the rows of $\Lambda^j$ accordingly and normalize its columns back to $\Delta^{m-1}$ to define convex combinations:

$$M^j = \mathrm{diag}(a^j)\Lambda^j \tag{10}$$

$$\hat{M}^j = M^j \mathrm{diag}(\mathbf{1}_m^\top M^j)^{-1} \tag{11}$$

We then interpolate embeddings and targets by taking $n$ convex combinations over $m$ examples:

$$\widetilde{Z}^j = Z^j \hat{M}^j \tag{12}$$

$$\widetilde{Y}^j = Y \hat{M}^j. \tag{13}$$

This is similar to (6),(7), but there is a different interpolated embedding matrix $\widetilde{Z}^j \in \mathbb{R}^{d \times n}$ as well as target matrix $\widetilde{Y}^j \in \mathbb{R}^{c \times n}$ per position, even though the original target matrix $Y$ is one. The total number of interpolated features and targets per mini-batch is now $nr$.

**Classifier** The classifier is now $g_W : \mathbb{R}^{d \times r} \to \mathbb{R}^{c \times r}$, maintaining the same spatial resolution as the embedding and generating one vector of predicted probabilities per spatial position. This is done by removing average pooling or any down-sampling operation. The interpolated embeddings $\widetilde{Z}^1, \ldots, \widetilde{Z}^r$ (12) are grouped by example into $\widetilde{\mathbf{z}}_1, \ldots, \widetilde{\mathbf{z}}_n \in \mathbb{R}^{d \times r}$, mapped by $g_W$ to predicted probabilities $\widetilde{\mathbf{p}}_1, \ldots, \widetilde{\mathbf{p}}_n \in \mathbb{R}^{c \times r}$ and grouped again by position into $\widetilde{P}^1, \ldots, \widetilde{P}^r \in \mathbb{R}^{c \times n}$.

In the simple case where the original classifier is linear, *i.e.* $W \in \mathbb{R}^{d \times c}$, it is seen as $1 \times 1$ convolution and applied densely to each column (feature) of $\widetilde{Z}^j$ for $j = 1, \ldots, r$.

**Loss** Finally, we learn parameters $\theta, W$ by minimizing the *weighted cross-entropy* $H(\widetilde{Y}^j, \widetilde{P}^j; s)$ of $\widetilde{P}^j$ relative to the interpolated targets $\widetilde{Y}^j$ again densely at each position $j$, where

$$H(Y, P; s) := -\mathbf{1}_c^\top (Y \odot \log(P)) s / (\mathbf{1}_n^\top s) \tag{14}$$

generalizes (1) and the weight vector is defined as $s = \mathbf{1}_m^\top M^j \in \mathbb{R}^n$. This is exactly the vector used to normalize the columns of $M^j$ in (11). The motivation is that the columns of $M^j$ are the original interpolation vectors weighted by attention: A small $\ell_1$ norm indicates that for the given position $j$, we are sampling from examples of low attention, hence the loss is to be discounted. The total number of loss terms per mini-batch is now $nr$.

# 4 Experiments

## 4.1 Setup

We use a mini-batch of size $b = 128$ examples in all experiments. Following manifold mixup [51], for every mini-batch, we apply MultiMix with probability $0.5$ or input mixup otherwise. For MultiMix, the default settings are given in subsection 4.6. We use PreActResnet-18 (R-18) [21] and WRN16-8 [63] as encoder on CIFAR-10 and CIFAR-100 datasets [29]; R-18 on TinyImagenet [60] (TI); and Resnet-50 (R-50) and ViT-S/16 [13] on ImageNet [44]. To better understand the effect of mixup in ViT, we evaluate MultiMix and Dense MultiMix on ImageNet without using strong augmentations like Auto-Augment [9], Rand-Augment [10], random erasing [67] and CutMix [62]. We reproduce TransMix [7] and TokenMix [35] using these settings.

## 4.2 Results: Image classification and robustness

**Image classification** In Table 2 we observe that MultiMix and Dense MultiMix already outperform SoTA on all datasets except CIFAR-10 with R-18, where they are on par with Co-Mixup. Dense MultiMix improves over vanilla MultiMix and its effect is complementary on all datasets. On TI for example, Dense MultiMix improves over MultiMix by 1.33% and SoTA by 1.59%. We provide additional analysis of the embedding space on 10 classes of CIFAR-100 in subsection 4.4.

| Dataset Network | Cifar-10 R-18 | W16-8 | Cifar-100 R-18 | W16-8 | TI R-18 |
|---|---|---|---|---|---|
| Baseline[†] | $95.41_{\pm0.02}$ | $94.93_{\pm0.06}$ | $76.69_{\pm0.26}$ | $78.80_{\pm0.55}$ | $56.49_{\pm0.21}$ |
| Manifold mixup [51][†] | $97.00_{\pm0.05}$ | $96.44_{\pm0.02}$ | $80.00_{\pm0.34}$ | $80.77_{\pm0.26}$ | $59.31_{\pm0.49}$ |
| PuzzleMix [26][†] | $97.04_{\pm0.04}$ | $\underline{97.00}_{\pm0.03}$ | $79.98_{\pm0.05}$ | $80.78_{\pm0.23}$ | $63.52_{\pm0.42}$ |
| Co-Mixup [25][†] | $\mathbf{97.10}_{\pm0.03}$ | $96.44_{\pm0.08}$ | $80.28_{\pm0.13}$ | $80.39_{\pm0.34}$ | $64.12_{\pm0.43}$ |
| AlignMixup [48][†] | $97.06_{\pm0.04}$ | $96.91_{\pm0.01}$ | $\underline{81.71}_{\pm0.07}$ | $\underline{81.24}_{\pm0.02}$ | $\underline{66.85}_{\pm0.07}$ |
| $\zeta$-Mixup [1][⋆] | $96.26_{\pm0.04}$ | $96.35_{\pm0.04}$ | $80.46_{\pm0.26}$ | $79.73_{\pm0.15}$ | $63.18_{\pm0.14}$ |
| MultiMix (ours) | $97.07_{\pm0.03}$ | $97.06_{\pm0.02}$ | $81.82_{\pm0.04}$ | $81.44_{\pm0.03}$ | $67.11_{\pm0.04}$ |
| Dense MultiMix (ours) | $97.09_{\pm0.02}$ | $\mathbf{97.09}_{\pm0.02}$ | $\mathbf{81.93}_{\pm0.04}$ | $\mathbf{81.77}_{\pm0.03}$ | $\mathbf{68.44}_{\pm0.05}$ |
| Gain | **-0.01** | **+0.09** | **+0.22** | **+0.53** | **+1.59** |

Table 2: *Image classification* on CIFAR-10/100 and TI (TinyImagenet). Mean and standard deviation of Top-1 accuracy (%) for 5 runs. R: PreActResnet, W: WRN. ⋆: reproduced, †: reported by AlignMixup, **Bold black**: best; Blue: second best; underline: best baseline. Gain: improvement over best baseline. Additional comparisons are in subsection A.2.

| Attack | FGSM | | | | | PGD | | | |
|---|---|---|---|---|---|---|---|---|---|
| Dataset | Cifar-10 | | Cifar-100 | | TI | Cifar-10 | | Cifar-100 | |
| Network | R-18 | W16-8 | R-18 | W16-8 | R-18 | R-18 | W16-8 | R-18 | W16-8 |
| Baseline[†] | $88.8_{\pm0.11}$ | $88.3_{\pm0.33}$ | $87.2_{\pm0.10}$ | $72.6_{\pm0.22}$ | $91.9_{\pm0.06}$ | $99.9_{\pm0.0}$ | $99.9_{\pm0.01}$ | $99.9_{\pm0.01}$ | $99.9_{\pm0.01}$ |
| Manifold mixup [51][†] | $76.9_{\pm0.14}$ | $76.0_{\pm0.04}$ | $80.2_{\pm0.06}$ | $56.3_{\pm0.10}$ | $89.3_{\pm0.06}$ | $97.2_{\pm0.01}$ | $98.4_{\pm0.03}$ | $99.6_{\pm0.01}$ | $98.4_{\pm0.03}$ |
| PuzzleMix [26][†] | $57.4_{\pm0.22}$ | $60.7_{\pm0.02}$ | $78.8_{\pm0.09}$ | $57.8_{\pm0.03}$ | $83.8_{\pm0.05}$ | $97.7_{\pm0.01}$ | $97.0_{\pm0.01}$ | $96.4_{\pm0.02}$ | $95.2_{\pm0.03}$ |
| Co-Mixup [25][†] | $60.1_{\pm0.05}$ | $58.8_{\pm0.10}$ | $77.5_{\pm0.02}$ | $56.5_{\pm0.04}$ | – | $97.5_{\pm0.02}$ | $\underline{96.1}_{\pm0.03}$ | $95.3_{\pm0.03}$ | $94.2_{\pm0.01}$ |
| AlignMixup [48][†] | $\underline{54.8}_{\pm0.03}$ | $\underline{56.0}_{\pm0.05}$ | $\underline{74.1}_{\pm0.04}$ | $\underline{55.0}_{\pm0.03}$ | $78.8_{\pm0.03}$ | $95.3_{\pm0.04}$ | $96.7_{\pm0.03}$ | $\underline{90.4}_{\pm0.01}$ | $\underline{92.1}_{\pm0.03}$ |
| $\zeta$-Mixup [1][⋆] | $72.8_{\pm0.23}$ | $67.3_{\pm0.24}$ | $75.3_{\pm0.21}$ | $68.0_{\pm0.21}$ | $\underline{84.7}_{\pm0.18}$ | $98.0_{\pm0.06}$ | $98.6_{\pm0.03}$ | $97.4_{\pm0.10}$ | $96.1_{\pm0.10}$ |
| MultiMix (ours) | $\mathbf{54.1}_{\pm0.09}$ | $55.3_{\pm0.04}$ | $73.8_{\pm0.04}$ | $54.5_{\pm0.01}$ | $77.5_{\pm0.01}$ | $94.2_{\pm0.04}$ | $94.8_{\pm0.01}$ | $90.0_{\pm0.01}$ | $91.6_{\pm0.01}$ |
| Dense MultiMix (ours) | $\mathbf{54.1}_{\pm0.01}$ | $\mathbf{53.3}_{\pm0.03}$ | $\mathbf{73.5}_{\pm0.03}$ | $\mathbf{52.9}_{\pm0.04}$ | $\mathbf{75.5}_{\pm0.04}$ | $\mathbf{92.9}_{\pm0.04}$ | $\mathbf{92.6}_{\pm0.01}$ | $\mathbf{88.6}_{\pm0.03}$ | $\mathbf{90.8}_{\pm0.01}$ |
| Gain | **+0.7** | **+2.7** | **+0.6** | **+2.1** | **+3.3** | **+2.4** | **+3.5** | **+1.4** | **+1.3** |

Table 4: *Robustness to FGSM & PGD attacks*. Mean and standard deviation of Top-1 error (%) for 5 runs: lower is better. ⋆: reproduced, †: reported by AlignMixup. **Bold black**: best; Blue: second best; underline: best baseline. Gain: reduction of error over best baseline. TI: TinyImagenet. R: PreActResnet, W: WRN. Comparison with additional baselines is given in subsection A.2.

In Table 3 we observe that on ImageNet with R-50, vanilla MultiMix outperforms all methods except AlignMixup. Dense MultiMix outperforms all SoTA with both R-50 and ViT-S/16, bringing an overall gain of 3% over the baseline with R-50 and 2.2% with ViT-S/16. The gain over AlignMixup with R-50 is small, but it is impressive that it comes with only linear interpolation. To better isolate the effect of each method, we reproduce TransMix [7] and TokenMix [35] with ViT-S/16 using their official code with our settings, *i.e.*, without strong regularizers like CutMix, Auto-Augment, Random-Augment *etc*. MultiMix is on par, while Dense MultiMix outperforms them by 1%.

**Training speed** Table 3 also shows the training speed as measured on NVIDIA V-100 GPU including forward and backward pass. The vanilla MultiMix has nearly the same speed with the baseline, bringing an accuracy gain of 2.49% with R-50. Dense MultiMix is slightly slower, increasing the gain to 3.10%. The inference speed is the same for all methods.

**Robustness to adversarial attacks** We follow the experimental settings of AlignMixup [48] and use $8/255$ $l_\infty$ $\epsilon$-ball for FGSM [16] and $4/255$ $l_\infty$ $\epsilon$-ball with step size $2/255$ for PGD [38] attack.

| Network Method | ResNet-50 Speed | Acc | ViT-S/16 Speed | Acc |
|---|---|---|---|---|
| Baseline[†] | 1.17 | 76.32 | 1.01 | 73.9 |
| Manifold mixup [51][†] | 1.15 | 77.50 | 0.97 | 74.2 |
| PuzzleMix [26][†] | 0.84 | 78.76 | 0.73 | 74.7 |
| Co-Mixup [25][†] | 0.62 | – | 0.57 | 74.9 |
| TransMix [7][⋆] | – | – | 1.01 | 75.1 |
| TokenMix [35][⋆] | – | – | 0.87 | $\underline{75.3}$ |
| AlignMixup [48][†] | 1.03 | $\underline{79.32}$ | – | – |
| MultiMix (ours) | 1.16 | 78.81 | 0.98 | 75.2 |
| Dense MultiMix (ours) | 0.95 | **79.42** | 0.88 | **76.1** |
| Gain | | **+0.1** | | **+1.2** |

Table 3: *Image classification and training speed* on ImageNet. Top-1 accuracy (%): higher is better. Speed: images/sec ($\times10^3$): higher is better. †: reported by AlignMixup; ⋆: reproduced. **Bold black**: best; Blue: second best; underline: best baseline. Gain: improvement over best baseline. Comparison with additional baselines is given in subsection A.2.

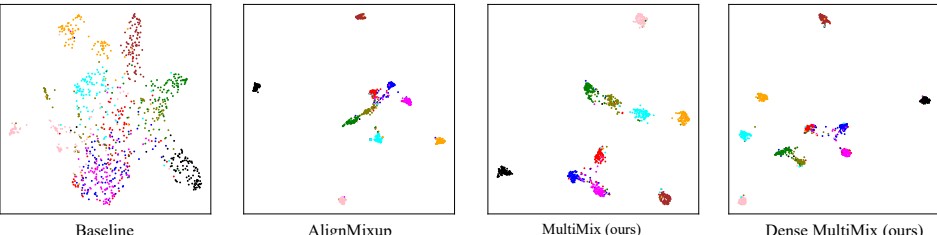

| Baseline | AlignMixup | MultiMix (ours) | Dense MultiMix (ours) |

Figure 3: *Embedding space visualization* for 100 test examples per class of 10 randomly chosen classes of CIFAR-100 with PreActResnet-18, using UMAP [39].

In Table 4 we observe that MultiMix is already more robust than SoTA on all datasets and settings. Dense MultiMix also increases the robustness and is complementary.

The overall gain is more impressive than in classification according to Table 2. For example, against the strong PGD attack on CIFAR-10 with W16-8, the SoTA Co-Mixup improves the baseline by 3.8% while Dense MultiMix improves it by 7.3%, which is double. MultiMix and Dense MultiMix outperform Co-Mixup and PuzzleMix by 3-6% in robustness on CIFAR-10, even though they are on-par on classification. There is also a significant gain over SoTA AlignMixup by 1-3% in robustness to FGSM on TinyImageNet and to the stronger PGD.

## 4.3 Results: Transfer learning to object detection

We evaluate the effect of mixup on the generalization ability of a pre-trained network to object detection as a downstream task. Following the settings of CutMix [62], we pre-train R-50 on ImageNet with mixup methods and use it as the backbone for SSD [36] with fine-tuning on Pascal VOC07+12 [15] and Faster-RCNN [43] with fine-tuning on MS-COCO [34].

In Table 5, we observe that, while MultiMix is slightly worse than AlignMixup on Pascal VOC07+12, Dense MultiMix brings improvements over the SoTA on both datasets and is still complementary. This is consistent with classification results. Dense MultiMix brings a gain of 0.8 mAP on Pascal VOC07+12 and 0.35 mAP on MS-COCO.

| DATASET | VOC07+12 | | MS-COCO | |
| DETECTOR | SSD | SPEED | FR-CNN | SPEED |
|---|---|---|---|---|
| Baseline[†] | 76.7 | 9.7 | 33.27 | 23.6 |
| Input mixup[†] | 76.6 | 9.5 | 34.18 | 22.9 |
| CutMix[†] | 77.6 | 9.4 | 35.16 | 23.2 |
| AlignMixup[†] | 78.4 | 8.9 | 35.84 | 20.4 |
| MultiMix (ours) | 77.9 | 9.6 | 35.93 | 23.2 |
| Dense MultiMix (ours) | **79.2** | 8.8 | **36.19** | 19.8 |
| Gain | **+0.8** | | **+0.35** | |

Table 5: *Transfer learning* to object detection. Mean average precision (mAP, %): higher is better. [†]: reported by AlignMixup. **Bold black**: best; Blue: second best; underline: best baseline. Gain: increase in mAP. Speed: images/sec: higher is better.

## 4.4 Analysis of the embedding space

**Qualitative analysis** We qualitatively analyze the embedding space on 10 CIFAR-100 classes in Figure 3. We observe that the quality of embeddings of the baseline is extremely poor with severely overlapping classes, which explains its poor performance on image classification. All mixup methods result in clearly better clustered and more uniformly spread classes. AlignMixup [48] yields five somewhat clustered classes and five moderately overlapping ones. Our best setting, *i.e.*, Dense MultiMix, results in five tightly clustered classes and another five somewhat overlapping but less than all competitors.

**Quantitative analysis** We also quantitatively assess the embedding space on the CIFAR-100 test set using alignment and uniformity [52]. *Alignment* measures the expected pairwise distance of examples in the same class. Lower alignment indicates that the classes are more tightly clustered. *Uniformity* measures the (log of the) expected pairwise similarity of all examples using a Gaussian kernel as a similarity function. Lower uniformity indicates that classes are more uniformly spread in the embedding space.

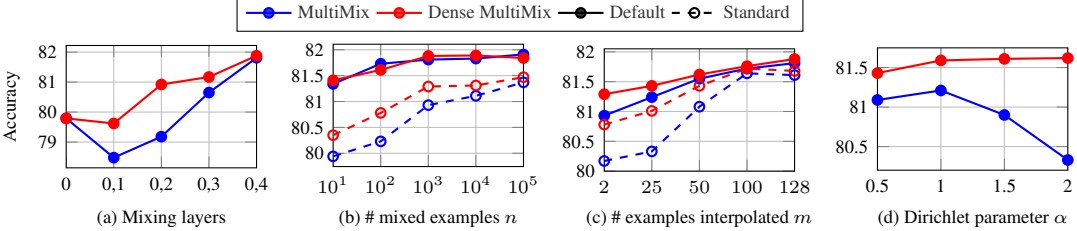

Figure 4: *Ablation study* on CIFAR-100 using R-18. (a) Interpolation layers (R-18 block; 0: input mixup). (b) Number $n$ of interpolated examples per mini-batch with $m = b$ (Default) and $m = 2$ (Standard). (c) Number $m$ of examples being interpolated, with $n = 1000$ (Default) and $n = 100$ (Standard). (d) Fixed value of Dirichlet parameter $\alpha$.

On CIFAR-100, we obtain alignment 3.02 for baseline, 2.04 for AlignMixup, 1.27 for MultiMix and 0.92 for Dense MultiMix. We also obtain uniformity -1.94 for the baseline, -2.38 for Align-Mixup [48], -4.77 for MultiMix and -5.68 for Dense MultiMix. These results validate the qualitative analysis of Figure 3.

## 4.5 Manifold intrusion analysis

Manifold intrusion [17] can occur when mixed examples are close to classes other than the ones being interpolated in the embedding space. To evaluate for manifold intrusion, we define the *intrusion distance* (ID) as the minimum distance of a mixed embedding to the clean embeddings of all classes except the ones being interpolated, averaged over a mini-batch: $\mathrm{ID}(\widetilde{Z}, Z) = \frac{1}{|\widetilde{Z}|} \sum_{\tilde{z} \in \widetilde{Z}} \min_{z \in Z} \|\tilde{z} - z\|^2$. Here, $\widetilde{Z}$ is the set of mixed embeddings in a mini-batch and $Z$ is the set of clean embeddings from all classes other than the ones being interpolated in the mini-batch. Intuitively, a larger $\mathrm{ID}(\widetilde{Z}, Z)$ denotes that mixed embeddings in $\widetilde{Z}$ are farther away from the manifold of other classes in $Z$, thereby preventing manifold intrusion.

Averaged over the training set of CIFAR-100 using Resnet-18, the intrusion distance is 0.46 for Input Mixup, 0.47 for Manifold Mixup, 0.45 for AlignMixup, 0.46 for MultiMix and 0.47 for Dense MultiMix. This is roughly the same for most SoTA mixup methods. This may be due to the fact that true data occupy only a tiny fraction of the embedding space, thus generated mixed examples lie in empty space between class-specific manifolds with high probability. The visualization in Figure 3 indeed shows that the embedding space is sparsely populated, even in two dimensions. This sparsity is expected to grow exponentially in the number of dimensions, which is in the order of $10^3$.

## 4.6 Ablations

All ablations are performed using R-18 on CIFAR-100. We study the effect of the layer where we interpolate, the number $n$ of generated examples per mini-batch, the number $m$ of examples being interpolated and the Dirichlet parameter $\alpha$. More ablations are given in the supplementary.

**Interpolation layer** For MultiMix, we use the entire network as the encoder $f_\theta$ by default, except for the last fully-connected layer, which we use as classifier $g_W$. Thus, we *interpolate embeddings* in the deepest layer by default. Here, we study the effect of different decompositions of the network $f = g_W \circ f_\theta$, such that interpolation takes place at a different layer. In Figure 4(a), we observe that mixing at the deeper layers of the network significantly improves performance. The same behavior is observed with Dense MultiMix, which validates our default choice.

It is interesting that the authors of input mixup [65] found that convex combinations of three or more examples in the input space with weights from the Dirichlet distribution do not bring further gain. This agrees with the finding of SuperMix [11] for four or more examples. Figure 4(a) suggests that further gain emerges when mixing in deeper layers.

**Number $n$ of generated examples per mini-batch** This is important since our aim is to increase the amount of data seen by the model, or at least part of the model. We observe from Figure 4(b) that accuracy increases overall with $n$ and saturates for $n \geq 1000$ for both variants of MultiMix. The improvement is more pronnounced when $m = 2$, which is standard for most mixup methods.

Our best solution, Dense MultiMix, works best at $n = 1000$ and $n = 10,000$. We choose $n = 1000$ as default, given also that the training cost increases with $n$. The training speed as a function of $n$ is given in the supplementary and is nearly constant for $n \leq 1000$.

**Number $m$ of examples being interpolated**   We vary $m$ between 2 (pairs) and $b = 128$ (entire mini-batch) by using $\Lambda' \in \mathbb{R}^{m \times n}$ drawn from Dirichlet along with combinations (subsets) over the mini-batch to obtain $\Lambda \in \mathbb{R}^{b \times n}$ with $m$ nonzero elements per column in (6),(7). We observe in Figure 4(c) that for both MultiMix and Dense MultiMix the performance increases with $m$. The improvement is more pronnounced when $n = 100$, which is similar to the standard setting ($n = b = 128$) of most mixup methods. Our choice of $m = b = 128$ brings an improvement of 1-1.8% over $m = 2$. We use this as our default setting.

**Dirichlet parameter $\alpha$**   Our default setting is to draw $\alpha$ uniformly at random from $[0.5, 2]$ for every interpolation vector (column of $\Lambda$). Here we study the effect of a fixed value of $\alpha$. In Figure 4(d), we observe that the best accuracy comes with $\alpha = 1$ for most MultiMix variants, corresponding to the uniform distribution over the convex hull of the mini-batch embeddings. However, all measurements are lower than the default $\alpha \sim U[0.5, 2]$. For example, from Table 2(a) (CIFAR-100, R-18), Dense MultiMix has accuracy 81.93, compared with 81.59 in Figure 4(d) for $\alpha = 1$.

## 5   Discussion

The take-home message of this work is that, instead of devising smarter and more complex interpolation functions in the input space or intermediate features, it is more beneficial to use MultiMix, even though its interpolation is only linear. In this work, we combine three elements:

1. Increase the number $n$ of generated mixed examples beyond the mini-batch size $b$.
2. Increase the number $m$ of examples being interpolated from $m = 2$ (pairs) to $m = b$.
3. Perform interpolation in the *embedding* space rather than the input space.

Figure 4 shows that all three elements are important in achieving SoTA performance and removing any one leads to sub-optimal results. We discuss their significance and interdependence here.

**Increasing the number $n$ of generated examples**   The *expected risk* is defined as an integral over the underlying continuous data distribution. Since that distribution is unknown, the integral is approximated by a finite sum, *i.e.*, the *empirical risk*. A better approximation is the *vicinal risk*, where a number of augmented examples is sampled from a distribution in the vicinity of each training example, thus *increasing the number of loss terms per training example*. Input mixup [65] is inspired by the vicinal risk. However, as a practical implementation, all mixup methods still generate $b$ mixed examples and thus incur $b$ loss terms for a mini-batch of size $b$. As discussed in section 2, previous works have used more loss terms than $b$ per mini-batch, but not for mixup.

Our hypothesis for the significance of element 1 is that more mixed examples, thus more loss terms by interpolation, provide a better approximation of the expected risk integral. Dense interpolation further increases the number of loss terms, thus further improving the quality of approximation.

**Increasing the number $m$ of examples being interpolated**   As discussed in section 2, previous works, starting from input mixup [65] have attempted to interpolate $m > 2$ examples in the input space by sampling $\lambda$ from Dirichlet or other distributions but have found the idea not effective for large $m$. Our finding is that element 2 becomes effective only by interpolating in the embedding space, that is, element 3.

**Interpolating in embedding space**   Element 3 is originally motivated by Manifold Mixup [51], where "interpolations in deeper hidden layers capture higher level information [64]." ACAI [4] explicitly studies interpolation in the latent space of an autoencoder to produce a smooth semantic warping effect in data space. This suggests that nearby points in the latent space are semantically similar, which in turn improves representation learning. Mixed examples generated by sampling in the embedding space lie on the learned manifold. We hypothesize that the learned manifold is a good surrogate of the true, unknown data manifold.

A natural extension of this work is to settings other than supervised classification. A limitation is that it is not straightforward to combine the sampling scheme of MultiMix with complex interpolation methods, unless they are fast to compute in the embedding space.

# 6 Acknowledgements

This work was in part supported by the ANR-19-CE23-0028 MEERQAT project and was performed using the HPC resources from GENCI-IDRIS Grant 2021 AD011012528.

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
