# A  More experiments

## A.1  More on setup

**Settings and hyperparameters**  We train MultiMix and Dense MultiMix with mixed examples only. We use a mini-batch of size $b = 128$ examples in all experiments. Following Manifold Mixup [51], for every mini-batch, we apply MultiMix with probability 0.5 or input mixup otherwise. For input mixup, we interpolate the standard $m = b$ pairs (5). For MultiMix, we use the entire network as the encoder $f_\theta$ by default, except for the last fully-connected layer, which we use as classifier $g_W$. We use $n = 1000$ tuples and draw a different $\alpha \sim U[0.5, 2.0]$ for each example from the Dirichlet distribution by default. For multi-GPU experiments, all training hyperparameters including $m$ and $n$ are per GPU.

For Dense MultiMix, the spatial resolution is $r = 4 \times 4 = 16$ on CIFAR-10/100 and $r = 7 \times 7 = 49$ on Imagenet by default. We obtain the attention map by (9) using GAP for vector $u$ and ReLU followed by $\ell_1$ normalization as non-linearity $h$ by default. To predict class probabilities and compute the loss densely, we use the classifier $g_W$ as $1 \times 1$ convolution by default; when interpolating at earlier layers, we follow the process described in subsection 3.3.

**CIFAR-10/100 training**  Following the experimental settings of AlignMixup [48], we train MultiMix and its variants using SGD for 2000 epochs using the same random seed as AlignMixup. We set the initial learning rate to 0.1 and decay it by a factor of 0.1 every 500 epochs. The momentum is set to 0.9 and the weight decay to 0.0001. We use a batch size $b = 128$ and train on a single NVIDIA RTX 2080 TI GPU for 10 hours.

**TinyImageNet training**  Following the experimental settings of PuzzleMix [26], we train Multi-Mix and its variants using SGD for 1200 epochs, using the same random seed as AlignMixup. We set the initial learning rate to 0.1 and decay it by a factor of 0.1 after 600 and 900 epochs. The momentum is set to 0.9 and the weight decay to 0.0001. We train on two NVIDIA RTX 2080 TI GPUs for 18 hours.

**ImageNet training**  Following the experimental settings of PuzzleMix [26], we train MultiMix and its variants using the same random seed as AlignMixup. We train R-50 using SGD with momentum 0.9 and weight decay 0.0001 and ViT-S/16 using AdamW with default parameters. The initial learning rate is set to 0.1 and 0.01, respectively. We decay the learning rate by 0.1 at 100 and 200 epochs. We train on 32 NVIDIA V100 GPUs for 20 hours.

**Tasks and metrics**  We use top-1 accuracy (%, higher is better) and top-1 error (%, lower is better) as evaluation metrics on *image classification* and *robustness to adversarial attacks* (subsection 4.2 and subsection A.2). Additional datasets and metrics are reported separately for *transfer learning to object detection* (subsection 4.3) and *out-of-distribution detection* (subsection A.3).

## A.2  More results: Classification and robustness

Using the experimental settings of subsection A.1, we extend Table 2, Table 3 and Table 4 of subsection 4.2 in Table 6, Table 7 and Table 8 respectively by comparing MultiMix and its variants with additional mixup methods. The additional methods are Input mixup [65], Cutmix [62], SaliencyMix [46], StyleMix [23], StyleCutMix [23], SuperMix [11] and $\zeta$-Mixup [1]. We reproduce $\zeta$-Mixup and SuperMix using the same settings. For SuperMix, we use the official code[1], which first trains the teacher network using clean examples and then the student using mixed. For fair comparison, we use the same network as the teacher and student models.

We observe that MultiMix and its variants outperform all the additional mixup methods on image classification. Furthermore, they are more robust to FGSM and PGD attacks as compared to these additional methods. The remaining observations in subsection 4.2 are still valid.

## A.3  More results: Reducing overconfidence

**Model calibration**  A standard way to evaluate over-confident predictions is to measure *model calibration*. We assess model calibration using MultiMix and Dense MultiMix on CIFAR-100. We

---

[1]https://github.com/alldbi/SuperMix

| Dataset | Cifar-10 | | Cifar-100 | | TI |
| Network | R-18 | W16-8 | R-18 | W16-8 | R-18 |
|---|---|---|---|---|---|
| Baseline[†] | 95.41±0.02 | 94.93±0.06 | 76.69±0.26 | 78.80±0.55 | 56.49±0.21 |
| Input mixup [65][†] | 95.98±0.10 | 96.18±0.06 | 79.39±0.40 | 80.16±0.1 | 56.60±0.16 |
| CutMix [62][†] | 96.79±0.04 | 96.48±0.04 | 80.56±0.09 | 80.25±0.41 | 56.87±0.39 |
| Manifold mixup [51][†] | 97.00±0.05 | 96.44±0.02 | 80.00±0.34 | 80.77±0.26 | 59.31±0.49 |
| PuzzleMix [26][†] | 97.04±0.04 | 97.00±0.03 | 79.98±0.05 | 80.78±0.23 | 63.52±0.42 |
| AugMix[⋆] [22] | 96.67±0.05 | – | 80.10±0.03 | – | – |
| Co-Mixup [25][†] | 97.10±0.03 | 96.44±0.08 | 80.28±0.13 | 80.39±0.34 | 64.12±0.43 |
| SaliencyMix [46][†] | 96.94±0.05 | 96.27±0.05 | 80.36±0.56 | 80.29±0.05 | 66.14±0.51 |
| StyleMix [23][†] | 96.25±0.04 | 96.27±0.04 | 80.01±0.79 | 79.77±0.17 | 63.88±0.27 |
| StyleCutMix [23][†] | 96.94±0.05 | 96.95±0.04 | 80.67±0.07 | 80.79±0.04 | 66.55±0.13 |
| SuperMix [11][‡] | 96.03±0.05 | 96.13±0.05 | 79.07±0.26 | 79.42±0.05 | 64.43±0.39 |
| AlignMixup [48][†] | 97.06±0.04 | 96.91±0.01 | 81.71±0.07 | 81.24±0.02 | 66.85±0.07 |
| $\zeta$-Mixup [1][⋆] | 96.26±0.04 | 96.35±0.04 | 80.46±0.26 | 79.73±0.15 | 63.18±0.14 |
| MultiMix (ours) | 97.07±0.03 | 97.06±0.02 | 81.82±0.04 | 81.44±0.03 | 67.11±0.04 |
| Dense MultiMix (ours) | 97.09±0.02 | 97.09±0.02 | 81.93±0.04 | 81.77±0.03 | 68.44±0.05 |
| Gain | -0.01 | +0.09 | +0.22 | +0.53 | +1.59 |

Table 6: *Image classification* on CIFAR-10/100 and TI (TinyImagenet). Top-1 accuracy (%): higher is better. R: PreActResnet, W: WRN. [⋆]: reproduced, [†]: reported by AlignMixup, [‡]: reproduced with same teacher and student model. **Bold black**: best; Blue: second best; underline: best baseline. Gain: improvement over best baseline.

| Network | ResNet-50 | | ViT-S/16 | |
| Method | Speed | Acc | Speed | Acc |
|---|---|---|---|---|
| Baseline[†] | 1.17 | 76.32 | 1.01 | 73.9 |
| Input mixup [65][†] | 1.14 | 77.42 | 0.99 | 74.1 |
| CutMix [62][†] | 1.16 | 78.60 | 0.99 | 74.2 |
| Manifold mixup [51][†] | 1.15 | 77.50 | 0.97 | 74.2 |
| PuzzleMix [26][†] | 0.84 | 78.76 | 0.73 | 74.7 |
| AugMix [22][⋆] | 1.12 | 77.70 | – | – |
| Co-Mixup [25][†] | 0.62 | – | 0.57 | 74.9 |
| SaliencyMix [46][†] | 1.14 | 78.74 | 0.96 | 74.8 |
| StyleMix [23][†] | 0.99 | 75.94 | 0.85 | 74.8 |
| StyleCutMix [23][†] | 0.76 | 77.29 | 0.71 | 74.9 |
| SuperMix [11][‡] | 0.92 | 77.60 | – | – |
| TransMix [7][⋆] | – | – | 1.01 | 75.1 |
| TokenMix [35][⋆] | – | – | 0.87 | 75.3 |
| AlignMixup [48][†] | 1.03 | 79.32 | – | – |
| MultiMix (ours) | 1.16 | 78.81 | 0.98 | 75.2 |
| Dense MultiMix (ours) | 0.95 | 79.42 | 0.88 | 76.1 |
| Gain | | +0.1 | | +1.2 |

Table 7: *Image classification and training speed* on ImageNet. Top-1 accuracy (%): higher is better. Speed: images/sec ($\times 10^3$): higher is better. [†]: reported by AlignMixup; [⋆]: reproduced; [‡]: reproduced with same teacher and student model. **Bold black**: best; Blue: second best; underline: best baseline. Gain: improvement over best baseline.

| ATTACK | FGSM | | | | | PGD | | | |
|---|---|---|---|---|---|---|---|---|---|
| DATASET | CIFAR-10 | | CIFAR-100 | | TI | CIFAR-10 | | CIFAR-100 | |
| NETWORK | R-18 | W16-8 | R-18 | W16-8 | R-18 | R-18 | W16-8 | R-18 | W16-8 |
| Baseline[†] | 88.8±0.11 | 88.3±0.33 | 87.2±0.10 | 72.6±0.22 | 91.9±0.06 | 99.9±0.0 | 99.9±0.01 | 99.9±0.01 | 99.9±0.01 |
| Input mixup [65][†] | 79.1±0.07 | 79.1±0.12 | 81.4±0.23 | 67.3±0.06 | 88.7±0.08 | 99.7±0.02 | 99.4±0.01 | 99.9±0.01 | 99.3±0.02 |
| CutMix [62][†] | 77.3±0.06 | 78.3±0.05 | 86.9±0.06 | 60.2±0.04 | 88.6±0.03 | 99.8±0.03 | 98.1±0.02 | 98.6±0.01 | 97.9±0.01 |
| Manifold mixup [51][†] | 76.9±0.14 | 76.0±0.04 | 80.2±0.06 | 56.3±0.10 | 89.3±0.06 | 97.2±0.01 | 98.4±0.03 | 99.6±0.01 | 98.4±0.03 |
| PuzzleMix [26][†] | 57.4±0.22 | 60.7±0.02 | 78.8±0.09 | 57.8±0.03 | 83.8±0.05 | 97.7±0.01 | 97.0±0.01 | 96.4±0.02 | 95.2±0.03 |
| AugMix [22][⋆] | 58.2±0.02 | – | 79.1±0.04 | – | – | 98.2±0.01 | – | 96.3±0.02 | – |
| Co-Mixup [25][†] | 60.1±0.05 | 58.8±0.10 | 77.5±0.02 | 56.5±0.04 | – | 97.5±0.02 | 96.1±0.03 | 95.3±0.03 | 94.2±0.01 |
| SaliencyMix [46][†] | 57.4±0.08 | 68.0±0.05 | 77.8±0.10 | 58.1±0.06 | 81.1±0.06 | 97.4±0.03 | 97.0±0.04 | 95.6±0.03 | 93.7±0.05 |
| StyleMix [23][†] | 80.0±0.23 | 71.2±0.21 | 80.6±0.15 | 68.2±0.17 | 85.1±0.16 | 98.1±0.09 | 97.5±0.07 | 98.3±0.09 | 98.3±0.09 |
| StyleCutMix [23][†] | 57.7±0.04 | 56.0±0.07 | 77.4±0.05 | 56.8±0.03 | 80.5±0.04 | 98.4±0.04 | 96.7±0.02 | 91.8±0.01 | 93.7±0.01 |
| SuperMix [11][‡] | 60.0±0.11 | 58.2±0.12 | 78.8±0.13 | 58.3±0.19 | 81.1±0.12 | 97.6±0.02 | 97.2±0.09 | 91.4±0.03 | 92.7±0.01 |
| AlignMixup [48][†] | 54.8±0.03 | 56.0±0.05 | 74.1±0.04 | 55.0±0.03 | 78.8±0.03 | 95.3±0.04 | 96.7±0.03 | 90.4±0.01 | 92.1±0.03 |
| ζ-Mixup [1][⋆] | 72.8±0.23 | 67.3±0.24 | 75.3±0.21 | 68.0±0.21 | 84.7±0.18 | 98.0±0.06 | 98.6±0.03 | 97.4±0.10 | 96.1±0.10 |
| MultiMix (ours) | 54.1±0.09 | 55.3±0.04 | 73.8±0.04 | 54.5±0.01 | 77.5±0.01 | 94.2±0.04 | 94.8±0.01 | 90.0±0.01 | 91.6±0.01 |
| Dense MultiMix (ours) | 54.1±0.01 | 53.3±0.03 | 73.5±0.03 | 52.9±0.04 | 75.5±0.04 | 92.9±0.04 | 92.6±0.01 | 88.6±0.03 | 90.8±0.01 |
| Gain | +0.7 | +2.7 | +0.6 | +2.1 | +3.3 | +2.4 | +3.5 | +1.4 | +1.3 |

Table 8: *Robustness to FGSM & PGD attacks*. Top-1 error (%): lower is better. [⋆]: reproduced, [†]: reported by AlignMixup. [‡]: reproduced, same teacher and student model. **Bold black**: best; Blue: second best; underline: best baseline. Gain: reduction of error over best baseline. TI: TinyImagenet. R: PreActResnet, W: WRN.

| METRIC | ECE | OE |
|---|---|---|
| Baseline | 10.25 | 1.11 |
| Input Mixup [65] | 18.50 | 1.42 |
| Manifold Mixup [51] | 18.41 | 0.79 |
| CutMix [62] | 7.60 | 1.05 |
| PuzzleMix [26] | 8.22 | 0.61 |
| Co-Mixup [25] | 5.83 | 0.55 |
| AlignMixup [48] | 5.78 | 0.41 |
| MultiMix (ours) | 5.63 | 0.39 |
| Dense MultiMix (ours) | **5.28** | **0.27** |

Table 9: *Model calibration* using R-18 on CIFAR-100. ECE: expected calibration error; OE: over-confidence error. Lower is better.

report *mean calibration error* (mCE) and *overconfidence error* (OE) in Table 9. MultiMix has lower error than all SoTA methods and Dense MultiMix even lower.

**Out-of-distribution detection** This is another standard way to evaluate over-confidence. Here, *in-distribution* (ID) are examples on which the network has been trained, and *out-of-distribution* (OOD) are examples drawn from any other distribution. Given a mixture of ID and OOD examples, the network should predict an ID example with high confidence and an OOD example with low confidence, *i.e.*, the confidence of the predicted class should be below a certain threshold.

Following AlignMixup [48], we compare MultiMix and its variants with SoTA methods trained using R-18 on CIFAR-100 as ID examples, while using LSUN [61], iSUN [56] and TI to draw OOD examples. We use detection accuracy, Area under ROC curve (AuROC) and Area under precision-recall curve (AuPR) as evaluation metrics. In Table 10, we observe that MultiMix and Dense MultiMix outperform SoTA on all datasets and metrics by a large margin. Although the gain of MultiMix and Dense MultiMix over SoTA mixup methods is small on image classification, they significantly reduce over-confident incorrect predictions and achieve superior performance on out-of-distribution detection.

## A.4 More results: Generalizing to unseen domains

We evaluate the ability of MultiMix and Dense MultiMix to generalize to unseen domains on the Office-Home dataset [50] under the open-domain setting, using the official settings of DAML [45].

| Task | Out-Of-Distribution Detection | | | | | | | | | | | |
|---|---|---|---|---|---|---|---|---|---|---|---|---|
| Dataset | LSUN (crop) | | | | iSUN | | | | TI (crop) | | | |
| Metric | Det Acc | AuROC | AuPR (ID) | AuPR (OOD) | Det Acc | AuROC | AuPR (ID) | AuPR (OOD) | Det Acc | AuROC | AuPR (ID) | AuPR (OOD) |
| Baseline[†] | 54.0 | 47.1 | 54.5 | 45.6 | 66.5 | 72.3 | 74.5 | 69.2 | 61.2 | 64.8 | 67.8 | 60.6 |
| Input mixup [65][†] | 57.5 | 59.3 | 61.4 | 55.2 | 59.6 | 63.0 | 60.2 | 63.4 | 58.7 | 62.8 | 63.0 | 62.1 |
| Cutmix [62][†] | 63.8 | 63.1 | 61.9 | 63.4 | 67.0 | 76.3 | 81.0 | 77.7 | 70.4 | 84.3 | 87.1 | 80.6 |
| Manifold mixup [51][†] | 58.9 | 60.3 | 57.8 | 59.5 | 64.7 | 73.1 | 80.7 | 76.0 | 67.4 | 69.9 | 69.3 | 70.5 |
| PuzzleMix [26][†] | 64.3 | 69.1 | 80.6 | 73.7 | 73.9 | 77.2 | 79.3 | 71.1 | 71.8 | 76.2 | 78.2 | 81.9 |
| AugMix [22][*] | 62.9 | 73.2 | 80.8 | 72.6 | 68.2 | 78.7 | 81.1 | 74.1 | 71.4 | 83.9 | 84.6 | 78.6 |
| Co-Mixup [25][†] | 70.4 | 75.6 | 82.3 | 70.3 | 68.6 | 80.1 | 82.5 | 75.4 | 71.5 | 84.8 | 86.1 | 80.5 |
| SaliencyMix [46][†] | 68.5 | 79.7 | 82.2 | 64.4 | 65.6 | 76.9 | 78.3 | 79.8 | 73.3 | 83.7 | 87.0 | 82.0 |
| StyleMix [23][†] | 62.3 | 64.2 | 70.9 | 63.9 | 61.6 | 68.4 | 67.6 | 60.3 | 67.8 | 73.9 | 71.5 | 78.4 |
| StyleCutMix [23][†] | 70.8 | 78.6 | 83.7 | 74.9 | 70.6 | 82.4 | 83.7 | 76.5 | 75.3 | 82.6 | 82.9 | 78.4 |
| SuperMix [11][‡] | 70.9 | 77.4 | 80.1 | 72.3 | 71.0 | 76.8 | 79.6 | 76.7 | 75.1 | 82.8 | 82.5 | 78.6 |
| AlignMixup [48][†] | 74.2 | 79.9 | 84.1 | 75.1 | 72.8 | 83.2 | 84.1 | 80.3 | 77.2 | 85.0 | 87.8 | 85.0 |
| ζ-Mixup [1][*] | 68.1 | 73.2 | 80.8 | 73.1 | 72.2 | 82.3 | 82.2 | 79.4 | 74.4 | 84.3 | 82.2 | 77.2 |
| MultiMix (ours) | 79.2 | 82.6 | 85.2 | 77.6 | 75.6 | 85.1 | 87.8 | 83.1 | 78.3 | 86.6 | 89.0 | **88.2** |
| Dense MultiMix (ours) | **80.8** | **84.3** | **85.9** | **78.0** | **76.8** | **85.4** | **88.0** | **84.6** | **81.4** | **89.0** | **90.8** | 88.0 |
| Gain | +6.6 | +4.4 | +1.8 | +2.9 | +2.9 | +2.2 | +3.9 | +4.3 | +4.2 | +4.0 | +3.0 | +3.2 |

Table 10: *Out-of-distribution detection* using R-18. Det Acc (detection accuracy), AuROC, AuPR (ID) and AuPR (OOD): higher is better. [*]: reproduced, [†]: reported by AlignMixup. [‡]: reproduced, same teacher and student model. **Bold black**: best; Blue: second best; underline: best baseline. Gain: increase in performance. TI: TinyImagenet.

| Domain | Clipart | Real-World | Product | Art |
|---|---|---|---|---|
| DAML [45] | 45.13 | 65.99 | 61.54 | 53.13 |
| MultiMix (ours) | 46.01 | 66.59 | 60.99 | 54.58 |
| Dense MultiMix (ours) | **46.32** | **66.87** | **62.28** | **56.01** |

Table 11: *Generalizing to unseen domains*. Image classification using R-18 on Office-Home dataset [50] under the open-domain setting, using the official settings of DAML [45]. Accuracy (%): higher is better.

Table 11 shows that, while both MultiMix and DAML use the Dirichlet distribution to sample interpolation weights, MultiMix and Dense MultiMix generalize to unseen domains better than DAML. We hypothesize this is due to sampling an arbitrarily large number of samples. In addition, Dense MultiMix brings significant gain, up to nearly 3%.

## A.5   More ablations

As in subsection 4.6, all ablations here are performed using R-18 on CIFAR-100.

**Mixup methods with dense loss**   In Table 6 we observe that dense interpolation and dense loss improve MultiMix. Here, we study the effect of the dense loss only when applied to SoTA mixup methods; dense interpolation is not straightforward or not applicable in general with other methods.

Given a mini-batch of $b$ examples, we follow the mixup strategy of the SoTA mixup methods to obtain the mixed embedding $\widetilde{Z}^j \in \mathbb{R}^{d \times b}$ for each spatial position $j = 1, \ldots, r$. Then, as discussed in subsection 3.3, we obtain the predicted class probabilities $\widetilde{P}^j \in \mathbb{R}^{c \times b}$ again for each $j = 1, \ldots, r$. Finally, we compute the cross-entropy loss $H(\widetilde{Y}, \widetilde{P}^j)$ (1) densely at each spatial position $j$, where the interpolated target label $\widetilde{Y} \in \mathbb{R}^{c \times b}$ is given by (4).

In Table 12, we observe that using a dense loss improves the performance of all SoTA mixup methods. The baseline improves by 1.4% accuracy ($76.76 \rightarrow 78.16$) and manifold mixup by 0.67% ($80.20 \rightarrow 80.87$). On average, we observe a gain of 0.7% brought by the dense loss. An exception is AlignMixup [48], which drops by 0.35% ($81.71 \rightarrow 81.36$). This may be due to the alignment

| METHOD | VANILLA | DENSE |
|---|---|---|
| Baseline | 76.76 | 78.16 |
| Input mixup [65] | 79.79 | 80.21 |
| CutMix [62] | 80.63 | 81.40 |
| Manifold mixup [51] | 80.20 | 80.87 |
| PuzzleMix [26] | 79.99 | 80.62 |
| Co-Mixup [25] | 80.19 | 80.84 |
| SaliencyMix [46] | 80.31 | 81.21 |
| StyleMix [23] | 79.96 | 80.76 |
| StyleCutMix [23] | 80.66 | 81.41 |
| SuperMix [11]‡ | 79.01 | 80.12 |
| AlignMixup [48] | 81.71 | 81.36 |
| MultiMix (ours)* | **81.81** | 81.84 |
| MultiMix (ours) | **81.81** | **81.88** |

Table 12: *The effect of dense loss*. Image classification on CIFAR-100 using R-18. Top-1 accuracy (%): higher is better. ‡: reproduced with same teacher and student model. *: Instead of Dense MultiMix, we only apply the loss densely.

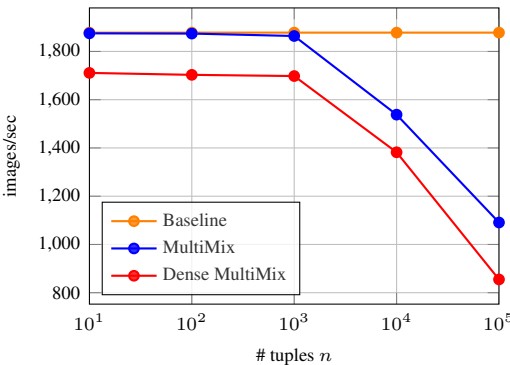

Figure 5: *Training speed* (images/sec) of MultiMix and its variants *vs*. number of tuples $n$ on CIFAR-100 using R-18. Measured on NVIDIA RTX 2080 TI GPU, including forward and backward pass.

process, whereby the interpolated dense embeddings are not very far from the original. MultiMix and Dense MultiMix still improve the state of the art under this setting.

**Training speed** In Figure 5, we analyze the training speed of MultiMix and Dense MultiMix as a function of number $n$ of interpolated examples. In terms of speed, MultiMix is on par with the baseline up to $n = 1000$, while bringing an accuracy gain of 5%. The best performing method—Dense MultiMix—is only slower by 10.6% at $n = 1000$ as compared to the baseline, which is arguably worth given the impressive 5.12% accuracy gain. Further increasing beyond $n > 1000$ brings a drop in training speed, due to computing $\Lambda$ and then using it to interpolate (6),(7). Because $n > 1000$ also brings little performance benefit according to Figure 4(b), we set $n = 1000$ as default for all MultiMix variants.

**Using a smaller batch size** We compare Input Mixup, Manifold Mixup and MultiMix for image classification using R-18 on CIFAR-100, with a batch size $b < 128$. By default, we use $n = 1000$ generated examples and $m = b$ examples being interpolated, following the same experimental settings described in subsection 4.1. As shown in Table 13, the increase of MultiMix accuracy with

| $m$ | 2 | 25 | 50 | 100 |
|---|---|---|---|---|
| Input Mixup [65] | 77.44 | 78.29 | 78.98 | 79.52 |
| Manifold Mixup [51] | 78.63 | 79.41 | 79.87 | 80.32 |
| MultiMix (ours) | **80.90** | **81.30** | **81.60** | **81.80** |

Table 13: *Effect of batch size* $m < 128$. Image classification using R-18 on CIFAR-100. Top-1 accuracy (%): higher is better.

| Method | $u$ | $h$ | Acc |
|---|---|---|---|
| Uniform | – | – | 81.33 |
| Attention (9) | CAM | softmax | 81.21 |
| | CAM | $\ell_1 \circ$ relu | 81.63 |
| | GAP | softmax | 81.78 |
| | GAP | $\ell_1 \circ$ relu | **81.88** |

Table 14: *Variants of spatial attention* in Dense MultiMix. Image classification on CIFAR-100 using R-18. Top-1 accuracy (%): higher is better. GAP: Global Average Pooling; CAM: Class Activation Maps [68]; $\ell_1 \circ$ relu: ReLU followed by $\ell_1$ normalization.

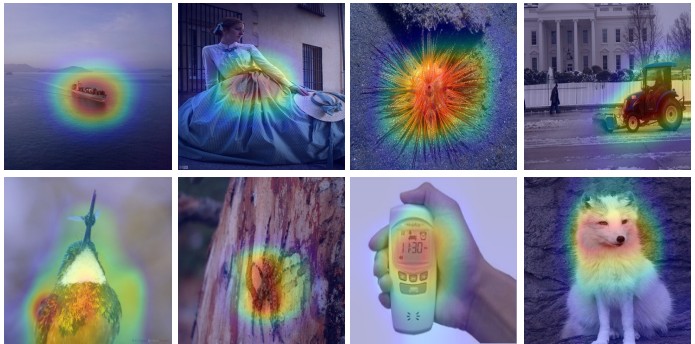

Figure 6: *Attention visualization.* Attention maps obtained by (9) with $u$ as GAP and $h$ as $\ell_1 \circ$ relu using Resnet-50 on the validation set of ImageNet. The attention localizes the complete or part of the object with high confidence.

increasing batch size $b$ is similar to the increase with the number $m$ of examples being interpolated, as observed in Figure 4(c). This is to be expected because, $m$ and $b$ are increasing together. An exhaustive hyper-parameter sweep could result in a different observation; currently, hyper-parameters are adjusted to the default choice $m = b = 128$. We also observe that the performance improvement of MultiMix over Input or Manifold Mixup is higher for smaller batch size. This may be due to the ability of MultiMix to draw from a larger pool of interpolated examples.

**Dense MultiMix: Spatial attention**   In subsection 3.3, we discuss different options for attention in dense MultiMix. In particular, no attention amounts to defining a uniform $a = \mathbf{1}_r/r$. Otherwise, $a$ is defined by (9). The vector $u$ can be defined as $u = \mathbf{z}\mathbf{1}_r/r$ by global average pooling (GAP) of $\mathbf{z}$, which is the default, or $u = Wy$ assuming a linear classifier with $W \in \mathbb{R}^{d \times c}$. The latter is similar to class activation mapping (CAM) [68], but here the current value of $W$ is used online while training. The non-linearity $h$ can be softmax or ReLU followed by $\ell_1$ normalization ($\ell_1 \circ$ relu), which is the default. Here, we study the affect of these options on the performance of dense Multimix.

In Table 14, we observe that using GAP for $u$ and $\ell_1 \circ$ relu as $h$ yields the best performance overall. Changing GAP to CAM or $\ell_1 \circ$ relu to softmax is inferior. The combination of CAM with softmax is the weakest, even weaker than uniform attention. CAM may fail because of using the non-optimal value of $W$ while training; softmax may fail because of being too selective. Compared to our best setting, uniform attention is clearly inferior, by nearly 0.6%. This validates that the use of spatial attention in dense MultiMix is clearly beneficial. Our intuition is that in the absence of dense targets, assuming the same target of the entire example at every spatial position naively implies that the object of interest is present everywhere, whereas spatial attention provides a better hint as to where the object may really be.

We validate this hypothesis in Figure 6, where we visualize the attention maps obtained using our best setting with $u$ as GAP and $h$ as $\ell_1 \circ$ relu. This shows that the attention map enables dense targets to focus on the object regions, which explains its superior performance.

**Dense MultiMix: Spatial resolution**   We study the effect of spatial resolution on dense MultiMix. By default, we use a resolution of $4 \times 4$ at the last residual block of R-18 on CIFAR-100. Here, we additionally investigate $1 \times 1$ (downsampling by average pooling with kernel size 4, same as GAP),

$2 \times 2$ (downsampling by average pooling with kernel size 2) and $8 \times 8$ (upsampling by using stride 1 in the last residual block). We measure accuracy 81.07% for spatial resolution $1 \times 1$, 81.43% for for $2 \times 2$, 81.88% for $4 \times 4$ and 80.83% for $8 \times 8$. We thus observe that performance improves with spatial resolution up to $4 \times 4$, which is the optimal, and then drops at $8 \times 8$. This drop may be due to assuming the same target at each spatial position. The resolution $8 \times 8$ is also more expensive computationally.