# OpenReview forum: "Embedding Space Interpolation Beyond Mini-Batch, Beyond Pairs and Beyond Examples"
_NeurIPS.cc/2023/Conference — NeurIPS 2023 poster_

### Official Review · Reviewer_SS7F · 2023-07-06

**Soundness:** 4 excellent
**Presentation:** 4 excellent
**Contribution:** 3 good
**Rating:** 6
**Confidence:** 3

**Summary:**

The paper introduces a novel data augmentation method called MultiMix, which aims to go beyond the limitations of existing mixup methods. The authors propose interpolating an arbitrarily large number of examples beyond the mini-batch size in the embedding space, rather than along linear segments between pairs of examples. They also extend the method to sequence data with Dense MultiMix, which densely interpolates features and target labels at each spatial location. The paper demonstrates that their proposed solutions outperform state-of-the-art mixup methods on various benchmarks.

*Contribution and novelty* The scientific contribution of the paper lies in the introduction of MultiMix and Dense MultiMix, which address the limitations of existing mixup methods. The authors propose novel approaches for generating an arbitrary number of interpolated examples beyond the mini-batch size, both in the embedding space and for sequence data. The empirical results, as described in the additional explanations, validate the effectiveness of these methods. Overall, this paper seems to improve the research frontier on data augmentation, but empirically and gives some new intuitive understanding of their results as their method leads to a more uniform distribution of embeddings within each class.  I do not see any clear weaknesses in the methods that may challenge the main claim of this work, but I have raised some issues in the questions section. Finally, since I have not read some of the papers that are cited in this work or other related papers in the literature, I defer the judgment about novelty of the ideas presented here to other reviewers.

**Strengths:**

- Novelty: The paper introduces a new data augmentation method, MultiMix, that extends the concept of mixup beyond the limitations of existing methods. The idea of interpolating in the embedding space and densely interpolating features and labels in sequence data is original and contributes to the field of data augmentation in machine learning.

- Empirical Results: The authors provide empirical evidence to support the effectiveness of their proposed solutions. They demonstrate significant improvements over state-of-the-art mixup methods on multiple benchmarks. The results validate the scientific contribution of the work and highlight its practical relevance.

- High quality of presentation of results: the presentation of the results follows a simple and logical flow, with main contributions and their difference to the previsous works clearly stated. The Methods, notations,  & results are all clearly stated and easy to understand for the reader.

- Intuitions about results: The paper provides some justifications & intuitions about  why their approach works. They do so by analyzing the embedding space. The authors show that the classes become more tightly clustered and uniformly spread over the embedding space, explaining the improved behavior of their solutions. This analysis adds depth to the scientific contribution of the work.

**Weaknesses:**

Since I have not read some of the papers that are cited in this work or other related papers in the literature, I defer the judgment about novelty of the ideas presented here to other reviewers. Besides that, I have raised a few issues in the questions section as they are not direct criticism of this work.

**Questions:**

I have the following questions from the authors:
- There is a lot of emphasis in the paper about the importance of number of loss terms and its importance for obtaining the results reported in this paper. However, one could achieve similarly high number of high loss terms by only incorporating several steps of pairwise mixup of the embeddings. This will have both random-ness in selection of pairs and value of $\lambda$ and will be equivalent to imposing a sparsity on $\lambda$'s that are sampled in Multimix. Is this point addressed in your evaluations or perhaps some cited works? In other words, what is exactly the benefit of having a dense $\lambda$ if doing mixup in the embedding space?
- It seems that the main benefits of the proposed approach is that it imposes new restrictions on the loss such that the solutions have uniform convex hulls per each input class. If authors agree with this statement, can they think of an alternative approach of achieving this? As a hypothetical experiment, can they think of an adding an explicit term in the loss that will encourage this uniform convex hulls, ie, some kind of a uniform convex hull regularisation term?

**Limitations:**

yes

---

> ### Author Rebuttal · Authors · 2023-08-09
>
> We appreciate the valuable feedback of R-SS7F. We address the concerns as follows:
>
> ### 1. Importance of using dense $\lambda$ in embedding space
>
> Using the description of our contributions presented in global thread **Author Rebuttal by Authors**, we elaborate on motivation here.
>
> ``R-SS7F - one could achieve similarly a high number of high loss terms by only incorporating several steps of pairwise mixup of the embeddings``
>
> This refers to using $m = 2$ rather than $m = b$ (element 2) for the same number $n$ of generated examples (element 1). In Fig. 4(c), we experiment with the configuration $m = 2, n = 1000$, which has a high number of loss terms, each obtained by mixing pairs, as in most mixup methods. This configuration results in 0.7\% lower accuracy compared to our default choice $m = b = 128, n = 1000$, thus empirically validating element 2.
>
> ``R-SS7F - what is exactly the benefit of having a dense $\lambda$ if doing mixup in the embedding space?``
>
> Using a dense $\lambda \in \Delta^{m-1}$ with $m = b$ sampled from the Dirichlet distribution *vs.* $\lambda \in [0,1]$ with $m = 2$ sampled from the Beta distribution refers to element 2.
>
> The motivation for element 2 is similar to the motivation of element 1, that is, to better approximate the expected risk integral by using a more uniform distribution for $\lambda$. The difference is best illustrated in Fig. 1: Even if we generate an arbitrarily large number $n$ of mixed examples with $m = 2$, these will always be constrained to linear segments between pairs of examples in the mini-batch, as shown in Fig. 1(a). By contrast, by using a dense $\lambda$ as in MultiMix, generated examples span the entire convex hull of the mini-batch examples, as shown in Fig. 1(b). Thus, we hypothesize that the expected risk integral is better approximated. This hypothesis motivating element 2 is validated in Fig. 4(c), as explained above.
>
> ### 2. Using a uniform convex hull regularizer
>
> This statement about a uniform convex hull per class does not describe what MultiMix is doing. The closest statement about MultiMix is best depicted in Fig. 1(b): We draw mixed examples from a distribution that is uniform (more precisely, Dirichlet with parameter $\alpha$ drawn uniformly at random) in the convex hull of the mini-batch examples, regardless of their class. Class labels are separately interpolated linearly as usual, see eq. (4).

---

> > ### Comment · Reviewer_SS7F · 2023-08-15
> >
> > I thank the authors for their clarifications, in particular for the explanation that the convex hull you sample from is not per class, but all data points. But this was not my question. Let me rephrase the question. Suppose that we fix a network architecture and get last hidden representations for vanilla $h_1,\dots, h_N$ and for network with MultiMix $h'_1, \dots , h'_N$. I was wondering if you can describe conceptually, how would $h_i$'s differ from $h'_i$'s? I then added my guess, that if you partition $h'_i$'s per class, they will form separated convex hulls.  While this is not directly a criticism, I have a hard time understanding why adding convex combinations of data-points with mixed classes will bring any benefits, which is why I posed the question.

---

> > > ### Author Response · Authors · 2023-08-17
> > > **Follow-up response to R-SS7F**
> > >
> > > Thank you for clarifying the question.
> > >
> > > ``can you describe conceptually, how would  $h_i$'s differ from  $h'_i$'s``
> > >
> > > We understand that you are referring to embeddings of clean examples, only with the network trained in two different ways. The embeddings $h'_i$ obtained by the network trained with MultiMix would form more tight clusters per class than the $h_i$ obtained without MultiMix. This is illustrated in Figure 3, MultiMix vs. Baseline, and quantified by alignment and uniformity measurements in paragraph "Quantitative analysis" in Section 4.4. However, there is no guarantee that class-specific convex hulls would be separated.
> > >
> > > ``why adding convex combinations of data points with mixed classes will bring any benefit``
> > >
> > > The most important properties that SoTA mixup methods focus on are robustness to corruption or adversarial attacks, the ability to detect out-of-distribution examples and model calibration. All mixup methods improve all these properties against the baseline, but they also improve model accuracy, by resulting in embeddings of clean examples that are more tightly clustered per class. This happens even though the loss function under mixup applies to mixed examples.
> > >
> > > Thulasidasan et al. [4*] claim that "improved calibration of a mixup classifier can be viewed as the classifier learning the true posteriors $P(Y|X)$ in the infinite data limit". In MultiMix, we increase the number $n$ of generated examples and also the number of loss terms beyond $b$. MultiMix is thus a better approximation of the expected risk integral. We understand that this is why MultiMix brings more improvement in all the properties above.
> > >
> > > [4*] Thulasidasan et al., On Mixup Training: Improved Calibration and Predictive Uncertainty for Deep Neural Networks, NeurIPS 2019
> > >
> > >
> > > We are also willing to clarify in case there are any further queries.

---

> > > > ### Comment · Reviewer_SS7F · 2023-08-17
> > > >
> > > > Thank you for the explanations. Now I think I have a better grasp on the main idea. This addresses all my previous comments.
> > > > However, upon reading respon I have a new question that may be critical to the main selling point of the paper.
> > > >
> > > > Given the similarity of MultiMix to existing methods that the empirical improvement over the existing methods are very small, it is very important to establish that these margins are statistically significant. Therefore, one can attribute the test accuracies reported in Tables 1-4 to the overfitting of hyper-parameters. Given the large number of combination of hyper-parameters as well as the small margins of improvement, the statistical significance reported in Tables 1-4 can be called into question.
> > > >
> > > > While I understand that doing a nested cross validation for all the models and datasets can be unreasonably burdensome. Perhaps as less burdensome suggestion, the authors do a nested cross-validation where hyperparameters are optimized over validation set, and then report test accuracy, for one of the smaller models and CIFAR100.  Otherwise, can the authors provide evidence and/or arguments that will hold up against statistical scrutiny?

---

> > > > > ### Author Response · Authors · 2023-08-20
> > > > > **Follow-up response to R-SS7F**
> > > > >
> > > > > Thank you for the question. We address the concern as follows:
> > > > >
> > > > > ``empirical improvement over the existing methods are very small``
> > > > >
> > > > > As mentioned in our previous response: "The most important metrics that SoTA mixup methods focus on is robustness to corruption or adversarial attacks, ability to detect out-of-distribution images and improve model calibration".
> > > > >
> > > > > In Section A.4 and Table 9 of the Appendix, we show that although the gain of MultiMix and Dense MultiMix over SoTA mixup methods is modest on image classification, they significantly reduce over-confident incorrect predictions and achieve superior performance on out-of-distribution detection, with gain ranging from 1.8\% to 6.6\%.
> > > > > On robustness to corruptions (Response to R-Xd45), we observe that both MultiMix and Dense MultiMix outperform Aug-Mix up to 4\% on CIFAR-100-C, and are also robust to corruptions on ImageNet-C.
> > > > >
> > > > > In Table 4, the gain in robustness to adversarial attacks ranges from 0.7 to 3.5\% accuracy. In Table 2, the gain in classification accuracy is 1.59\% on TinyImagenet.
> > > > >
> > > > > Please see also our response "3. Additional empirical evidence" to R-Xd45.
> > > > >
> > > > > ``it is very important to establish that these margins are statistically significant``
> > > > >
> > > > > Most of the improvements in Table 2 (image classification) and Table 4 (robustness to adversarial attacks) are statistically significant, as indicated by the standard deviations measured over five runs for each experiment (L227-228). For example, Table 2, CIFAR-100 using W16-8, mean accuracy gain 0.53\% with standard deviation 0.03\%. Multiple runs are not feasible on large datasets like ImageNet (Table 3) and MS-COCO (Table 5).
> > > > >
> > > > > ``one can attribute the test accuracies reported in Tables 1-4 to the overfitting of hyper-parameters.``
> > > > >
> > > > > The main hyperparameters of MultiMix and Dense MultiMix are ablated on CIFAR-100 using ResNet-18, as shown in Figure 4, and the default choices are then used on all remaining experiments. These choices are guided by computational constraints in addition to performance.
> > > > >
> > > > > For example, the mixing layer (Fig. 4a) is chosen as the last one, which is the most efficient because it reduces to a single matrix multiplication (eq. 6) and requires no forward passes for the mixed examples. The number $n$ of generated examples (Fig. 4b) is chosen as $10^3$, which is the highest possible before the image throughput reduces significantly, as shown in Figure 5. The number $m$ of examples being interpolated (Fig. 4c) is chosen as the batch size $b$, which is the simplest because any choice $m < b$ would need to additionally take subsets over the mini-batch (L320-322). The Dirichlet hyperparameter $\alpha$ (Fig. 4d) is chosen to be drawn uniformly at random from $[0.5,2]$ (L327-328), which is the interval where accuracy is roughly within 1\% of the optimal.
> > > > >
> > > > > For the training hyper-parameters, we follow the experimental setting of manifold mixup [36] for CIFAR-10/100 (L219-220; L470-471 - Section A.1), PuzzleMix [16] for TinyImagenet and Imagenet (L487; L492 - Section A.1). For Pascal-VOC and COCO, we follow CutMix [45] (L269-270). Thus, there has been no particular effort to overfit any hyperparameter on any dataset.
> > > > >
> > > > > ``authors do a nested cross-validation where hyperparameters are optimized over validation set, and then report test accuracy``
> > > > >
> > > > > Performing a nested cross-validation for the main hyperparameters of MultiMix and Dense MultiMix would imply repeating the process and making different choices for each dataset and network; while for other methods we naturally use the defaults provided by their authors, which are the same for all datasets and networks. This would be an unfair comparison.

---

> > > > > > ### Comment · Reviewer_SS7F · 2023-08-20
> > > > > >
> > > > > > I thank the authors for the further clarifications. Specially for the elaborate description of the way that hyper parameters were chosen. Here are my further comments:
> > > > > >
> > > > > > > Performing a nested cross-validation ... This would be an unfair comparison.
> > > > > >
> > > > > > A nested cross-validation score is not to be compared directly with SoTA scores. Rather, it evaluates if your optimal choice of hyper parameters is overly sensitive to choice data. More specifically, if the best configuration chosen over randomly split validation set performs poorly on the remaining test set, it signals a high risk of hyper parameter over fitting. Reporting nested CV and default model scores on some small model and small dataset will go a long way in addressing this problem. While I do not doubt that there is a good chance that there is not such a wide gap between the two, the current experiments do not rule out such a possibility. I will update my score to borderline accept to reflect my current assessment.

---

> > > > > > > ### Author Response · Authors · 2023-08-21
> > > > > > > **Follow-up response to R-SS7F**
> > > > > > >
> > > > > > > We understood that the discussion on overfitting of hyper-parameters implied overfitting to a particular dataset/network, hence we would need to repeat for different datasets/networks.
> > > > > > >
> > > > > > > We apologize for our misunderstanding and thank you for the clarification. Based on the reviewer's suggestion, we perform a nested cross-validation to study the effect of the number $n$ of generated examples in MultiMix. We divide the official training set of CIFAR-100 into "train set" and "test set", which comprise the outer loop. The "train set" is again split into "train subset" and "validation subset", comprising the inner loop on which the hyper-parameters are tuned. We report the mean accuracy on the "validation subset" in the Table below.
> > > > > > >
> > > > > > > |      | mean accuracy |
> > > > > > > |------|---------|
> > > > > > > | $n=10$   | 92.19   |
> > > > > > > | $n=100$  | 92.61   |
> > > > > > > | $n=1000$ | 93.43   |
> > > > > > >
> > > > > > > We observe that on the "validation subset", $n=1000$ results in the best overall performance. For the sake of comparison, we also report the performance on the "test set" and observe that the mean accuracy of $n=10$ is 87.43, $n=100$ is 88.08 and $n=1000$ is 88.73, which validates our observation in Figure 4(b).
> > > > > > >
> > > > > > > We thank the reviewer for the suggestion and shall add the experiments of nested cross-validation for other hyper-parameters in the camera-ready version of the paper.

---

> > > > > > > > ### Comment · Reviewer_SS7F · 2023-08-21
> > > > > > > >
> > > > > > > > I thank the authors for taking the time to add this crucial step to substantiate their results. I recommend adding a similar report for other hyper parameters. In particular case of $\alpha$, since it's chosen from an interval, it makes more sense to report the table for $\alpha$ being chosen from various ranges, rather than a fixed alpha.
> > > > > > > > Assuming that authors will add these validations, I have no further objections and thus will increase my score to weak accept.

---

> > > > > > > > > ### Author Response · Authors · 2023-08-22
> > > > > > > > > **Thank you R-SS7F**
> > > > > > > > >
> > > > > > > > > We sincerely thank R-SS7F for their time and effort in evaluating our work. We appreciate the insightful comments and interesting discussions, and for raising the score to "Weak Accept". We shall add all the additional experiments to the camera-ready version of the paper.

---

### Official Review · Reviewer_6vc1 · 2023-07-09

**Soundness:** 3 good
**Presentation:** 3 good
**Contribution:** 2 fair
**Rating:** 5
**Confidence:** 4

**Summary:**

This paper considers a generalized form of Manifold mixup by expanding the concept of a pair-wise interpolation method to the multi-sample-based interpolation on the entire convex hull. The proposed methods called MultiMix and Dense MultiMix show the best performance over prior mixup-based methods for the various classification benchmarks. Also, they are empirically shown to have better robustness than prior methods.


**Strengths:**

$\textbf{Strength 1:}$ The main strength of this paper is the suggestion of the general form of the prior mixup-based methods to cover the entire convex hull which can be formed by a given set of training samples. Because the power of the mixup methods originated from the interpolation of the space, i.e., either input or embedding, in-between different samples, expanding the concept of mixup into the entire convex hull of the given samples are clear and convincing direction.

$\textbf{Strength 2:}$ The paper is well-written and easy to understand. To be specific, Fig. 1 is simple yet effective to understand the main strength of the proposed method. Also, the formulations in the paper are quite neat and well-organized.

$\textbf{Strength 3:}$ The experiments are done for various kinds of architectures ranging from the ResNet family to ViT, and benchmarks including CIFAR and ImageNet cases.


**Weaknesses:**

$\textbf{Weakness 1:}$ The most crucial weakness of this paper is the novelty of the proposed method. Specifically, I admit that the expansion of the sample interpolation method to the generalized form, i.e., MultiMix, is convincing and the authors show a well-organized formulation of it, but I also feel that the expansion of the concept is quite easy to be anticipated from the prior Mixup methods. For example, in the original Mixup paper, the authors already tried to utilize multiple samples by weighting them with a simplex coefficient in the input space. Although it is reported not to show a further gain, it is hard to find a noticeable difference between the `trial' of the original Mixup paper and the proposed method except for the target space, i.e., input for Mixup vs. embedding space for MultiMix. When focusing on the difference, i.e., mixing on the embedding space, it is natural to ask for the reason why the multi-sample-based mixup becomes effective in the embedding space rather than the input space. However, I cannot find a deep insight into the phenomenon.

$\textbf{Weakness 2:}$ A more precise comparison in the Related Work part would be beneficial to highlight the significance of the proposed method. To be specific, in Related Work, a bunch of works are grouped into their attributes and briefly mentioned. The authors drop the detailed explanation of the prior works with a suggested survey paper but I am sure that the authors must provide a compact but clear explanation of the related works. As a simple suggestion, mentioning the names of prior mixup methods along with short descriptions would make it more clear. When a set of important baselines including AlignMix, Co-Mixup, PuzzleMix, and $\zeta$-Mixup are called with names in the Related Work part, readers can easily match the name of the priors appeared in Experiments and the descriptions in Related Work.

$\textbf{Weakness 3:}$ The metrics in 4.4 Quantitative analysis, i.e., 'Alignment' and 'Uniformity' do not fully represent the quality of the embedding space. These measurements can evaluate the similarity of embedded features of intra-class samples but do not capture the separation between different classes, i.e., the separability of inter-class distributions.

$textbf{Weakness 4:}$ Manifold intrusion is the main challenge of mixup-based methods. However, I cannot find any analysis for the effect of manifold intrusion of MultiMix. The experimental gains empirically show that the effect of manifold intrusion is minimal in practice. However, further discussion is needed on the effect of manifold intrusion of the multi-sample-based mixup approaches. I wonder if the chance of the intrusion might be higher than the pair-wise mixup of prior works.

**Questions:**

$\textbf{Question 1:}$ Would you provide a further explanation for the reason why the multi-sample mixup becomes when the interpolation is done in the deeper layer? Again, I felt that the main significance of this work is the finding of the effectiveness of multi-sample-based mixup on deeper layers. As mentioned in this paper, the authors of Mixup and SuperMix had tried to interpolate multiple samples but they are limited in the input space which leads to minimal performance gain. If the novelty becomes more clear and strong after the rebuttal, I will change the rating of the paper.

$\textbf{Question 2: }$ According to Weakness 3, the metrics used in Section 4.4 seem not to capture the separability between different classes. I think that when the 'Alignment' and 'Uniformity' values are divided into the similarities between different classes, they eventually consider the compactness and separability of the embedding. To be specific, when considering class $k$, we can measure the distance between the per-class averaged feature vector of class $k$ and the per-class averaged feature of the nearest interfering class. When we divide `Alignment' with the distance to the nearest different class, then the value will consider both the similarity between intra-class features and the dissimilarity between inter-class features. What happens when we compute the modified 'Alignment' and 'Uniformity'?

$\textbf{Question 3: }$ Would you provide a computational cost of the proposed methods and the comparison with other baselines? The results of the training speed guarantee that the proposed method does not suffer from further latency in practice, but I wonder if MultiMix and Dense MultiMix show a similar level of computational cost to the baselines.

$\textbf{Question 4: }$ Would you provide a discussion of the thought of manifold intrusion of the proposed methods? I imagine that the multi-sample-based mixup tries to cover the entire convex hull so it probably causes manifold intrusion more frequently.

**Limitations:**

Weakness 1 and Question 1 are for pointing out the limitation of the proposed work in view of novelty.

Also, I provide a suggestion for improving Related Work in Weakness 2.

As better metrics for quantitative analysis in section 4.4, I provide the modified measurements in Weakness 3 and Question 2.

For emphasizing the computational efficiency of MultiMix and Dense MultiMix, providing the computational costs would be beneficial (as aforementioned in Question 3).

For the manifold intrusion issue, I described the limitation and the corresponding question in Weakness 4 and Question 4.

---

> ### Author Rebuttal · Authors · 2023-08-09
>
> We appreciate the valuable feedback of R-6vc1. We address the concerns as follows:
> ### 1. Why multi-sample-based mixup is effective
> Using the description of our contributions presented in global thread **Author Rebuttal by Authors**, we elaborate on motivation here.
>
> ``R-6vc1 - why the multi-sample-based mixup becomes effective in the embedding space rather than the input space``
>
> Our motivation for element 3 i.e. interpolating in the embedding space, is twofold: First, generating $n$ samples (element 1) is computationally expensive when performed in the input space or in shallow layers; generating arbitrarily many samples without significantly increasing the cost is only feasible in the embedding space (L49-52, Fig. 5). Second, *"interpolation in the latent or embedding space is equivalent to interpolating along a manifold in the input space [36]''* (L27-28). We elaborate on this second motivation here.
>
> Interpolation in the embedding space is originally motivated by Manifold Mixup [36], which *"leverages semantic interpolations''*, In particular, it *"leverages interpolations in deeper hidden layers, which capture higher-level information
> (Zeiler \& Fergus, 2013) to provide additional training signal.''* Indeed, all nonlinear interpolation mechanisms proposed for mixup [45, 16, 15, 33, 3] can be thought of as performing a linear interpolation on a hand-engineered nonlinear manifold. In this sense, Manifold Mixup performs linear interpolation on the manifold learned from the data. *Adversarially constrained autoencoder interpolation* (ACAI) [3*] explicitly studies interpolation in the latent space of an autoencoder to produce a smooth semantic warping effect in data space. This suggests that nearby points in the latent space are semantically similar, which in turn improves representation learning.
>
> Based on the above, mixed examples generated by sampling in the embedding space across different classes lie on the learned manifold. We hypothesize that the learned manifold is a good surrogate of the true, unknown data manifold, which should improve the accuracy and robustness of the model. We also hypothesize that using the learned manifold becomes more important when interpolating more examples. This hypothesis motivating element 3 is validated in Fig. 4(a), where accuracy increases continuously as we shift interpolation to the deepest layer.
>
> [3*] Berthelot *et al.*, Understanding and improving interpolation in autoencoders via an adversarial regularizer. ICLR 2019.
>
> ### 2.  Metrics to measure class separability
> Thanks for the interesting suggestion. We formulate the modified alignment metric as
>
> $$\frac{\sum_{(x, x') \sim p_{\text{pos}}} ||f(x) - f(x') ||^2_2}{\sum_{(x, x') \sim p_{\text{neg}}} ||f(x) - f(x')||^2_2}.$$
>
> Here, $p_\text{pos}$ denotes the probability distribution of positive pairs (of the same class) and $p_\text{neg}$ the distribution of negative pairs (of different classes).
>
> We compare different methods by this modified metric on CIFAR-100 using Resnet-18. We obtain 0.63 for baseline, 0.44 for AlignMixup, 0.39 for MultiMix and 0.35 for Dense MultiMix. This shows that along with having small intra-class distance, MultiMix and Dense MultiMix have higher inter-class separability. We shall add this analysis to the final paper.
>
> ### 3. Manifold intrusion
> Manifold intrusion can occur when mixed examples are close to classes other than the ones being interpolated in the embedding space. To evaluate for manifold intrusion, we compute the *intrusion distance*, that is, the minimum distance between a mixed embedding and the clean embeddings of all classes except the ones being interpolated. This is formulated as
>
> $$\text{ID}(\tilde{Z},Z) = \frac{1}{|\tilde{Z} |}\sum_{\tilde{z} \in \tilde{Z}} \min_{z \in Z} ||\tilde{z} - z||^2.$$
>
> Here, $\tilde{Z}$ denotes a set of mixed embeddings in a mini-batch and $Z$ denotes the set of clean embeddings from all classes other than the ones being interpolated in the mini-batch. Intuitively, a larger $\text{ID}(\tilde{Z},Z)$ denotes that mixed embeddings in $\tilde{Z}$ are farther away from the manifold of other classes in $Z$, thereby preventing manifold intrusion.
>
> We compare different methods by this proposed metric on CIFAR-100 using Resnet-18. Averaged over the training set, the intrusion distance is 0.46 for Input Mixup, 0.47 for Manifold Mixup, 0.45 for AlignMixup, 0.46 for MultiMix and 0.47 for Dense MultiMix. This means that most SoTA mixup methods have roughly the same intrusion distance, thus MultiMix and Dense MultiMix do not suffer from a higher manifold intrusion than other methods.
>
> We thus validate the reviewer's observation that manifold intrusion is minimal in practice. This may be due to the fact that true data occupy only a tiny fraction of the embedding space, thus generated mixed examples may lie in empty space between class-specific manifolds with high probability. The visualization of Fig. 3 indeed shows that the embedding space is sparsely populated, even in two dimensions. This sparsity is expected to grow exponentially in the number of dimensions, which is in the order of $10^3$.
>
> ### 4. Computational Cost
> We compare the training speed of MultiMix and Dense MultiMix in Section 4.2 under "Training speed" (L243-249). We also ablate the training speed of MultiMix and its variants $vs.$ the number of tuples $n$ on CIFAR100 using R-18 in Section A.4 (Appendix) under "Training speed" (L543-550). In summary, MultiMix has nearly the same speed with the baseline, whereas Dense MultiMix is only slightly slower.
>
> ### 5.  Elaborating on related works
> Thanks for this suggestion. We organized related works based on the fact that most SoTA mixup methods focus on defining the interpolation operation, whereas we focus on the number and size of tuples used per mini-batch, as highlighted in Table 1. In the final paper, we shall provide names and discuss specific mixup methods that we have used for comparison in our experiments.

---

> > ### Comment · Reviewer_6vc1 · 2023-08-17
> >
> > Dear Authors,
> >
> > I really appreciate your additional efforts in providing the response.
> > - For the first issue about the layer where the mixup takes place, I am quite well convinced that the proposed method selected the last layer, where the choice is supported by the fact that many mixup-based methods are consistently showing the advantages of deeper layers. Also, thanks to the clarification of your contribution beyond the previous 'trials' by the prior works to utilize multiple samples, i.e., $m>3$.
> > - For the issues about '2. Metrics to measure the class separarity' and '3. Manifold intrusion', I believe that my concerns are well relieved by the additional results. I really thank you for your efforts. If they are added to your paper, I am pretty sure that your work will be more concrete.
> > - For the '4. Computational cost', I hoped to see the equation-based presentation of the computations in fact. However, I agree that the extensive simulations for training speed empirically guarantee the moderate computational overhead of your methods.
> > - For the '5. Elaborating on related works', I thank you for your clarification.
> >
> > As a consequence, I will increase my rating to 'Borderline Accept'. Again, I greatly appreciate your valuable feedback.
> > Sincerely,
> >
> > Reviewer 6vc1

---

> > > ### Author Response · Authors · 2023-08-17
> > > **Thank you R-6vc1**
> > >
> > > We sincerely thank R-6vc1 for their time and effort in evaluating our work, for the insightful comments, and for raising the score to "Borderline Accept". We shall add all experiments and clarifications of the rebuttal to the camera-ready version of the paper.

---

### Official Review · Reviewer_Hzfq · 2023-07-15

**Soundness:** 2 fair
**Presentation:** 2 fair
**Contribution:** 2 fair
**Rating:** 4
**Confidence:** 4

**Summary:**

The paper aims to increase the performance and robustness of deep learning model by increasing sample diversity. The authors propose data augmentation techniques which interpolate samples from a convex hull. The experimental evaluations show that the algorithm outperforms existing techniques.

**I have read the rebuttals. Please see the comments below for the discussion.**

**Strengths:**

1. Intuitive idea: the use of convex hull is natural and simple.
2. Applicability: the algorithm can be generally applied in many settings.

**Weaknesses:**

1. Presentation: the presentation of the paper needs significant improvement. For example, the abstract should succinctly summarizes the key ideas of the paper.
2. Novelty: sampling from a convex hull is not novel. The use of Dirichlet distribution has also been done in many other works. Could you please compare your work with the work by Shu et al. ?





[1] Shu, Y., Cao, Z., Wang, C., Wang, J., & Long, M. (2021). Open domain generalization with domain-augmented meta-learning. In Proceedings of the IEEE/CVF conference on computer vision and pattern recognition (pp. 9624-9633).

**Questions:**

1. It would be nice to see the analysis on larger/multi-modal models. I am wondering if the gain will be more significant since the sampling diversity will be crucial there. For example, could you conduct experiments on larger Vision Transformer models or language-augmented vision models such as CLIP?
2. In the ablation study, the performance seems to be sensitive to the hyper-parameters selected, especially since the performance difference from the second best algorithm is between 1-2%. Could you expand more on how the parameters are chosen for each experiment?

**Limitations:**

The authors discuss the limitation of the algorithm in the concluding paragraph.

---

> ### Author Rebuttal · Authors · 2023-08-09
>
> We appreciate the valuable feedback of Reviewer Hzfq. We address the concerns as follows:
>
> ### 1. Presentation
> We would like to point out that the other reviewers appreciate the writing and quality of the presentation. R-Xd45 and R-SS7F rate presentation as *excellent*. R-6vc1 acknowledges that The paper is *well-written* and *easy to understand*,  the formulations in the paper are *quite neat and well-organized*. R-SS7F says *High quality of presentation* of results: the presentation follows a *simple and logical flow*,  methods, notations, \& results are *clearly stated and easy to understand* for the reader.''
>
> We thank the reviewer for the suggestion on the abstract. For completeness, we discuss our motivation, the drawback of prior mixup methods, our two contributions--MultiMix and Dense MultiMix--as well as their performance on different benchmarks, all in 22 lines. We kindly ask the reviewer to help us on how can we further make our abstract more succinct without losing critical information. We believe that without the necessary background, the abstract might be difficult to follow. We also kindly ask the reviewer to point out any other examples where the presentation may be improved.
>
> ### 2. Novelty of using Dirichlet distribution
> We never claim that sampling the interpolation weights from Dirichlet distribution is our contribution (L81-90). In fact, in Table 1 and L111-112, we mention that SuperMix also uses a Dirichlet distribution over not more than 3 examples in practice.
>
> ### 3. Comparison with DAML
> Thanks for sharing the work by Shu *et al.*. We use the official settings of DAML and compare MultiMix and Dense MultiMix on the Office-Home dataset under the open-domain setting using Resnet-18 as the backbone. We report the accuracy in the table below:
>
> | Domain               | Clipart | Real-World | Product | Art    |
> |----------------------|---------|------------|---------|--------|
> | DAML                 | 45.13   | 65.99      | 61.54   | 53.13  |
> | MultiMix (ours)      | 46.01   | 66.59      | 60.99   | 54.58  |
> | Dense MultiMix (ours)| 46.32   | 66.87      | 62.28   | 56.01  |
>
> While both MultiMix and DAML sample from the convex hull and use the Dirichlet distribution to sample interpolation weights, MultiMix and Dense MultiMix generalize to unseen domains better than DAML, due to sampling an arbitrarily large number of samples. In addition, Dense MultiMix brings significant gain, up to nearly 3\%.
>
> ### 4. Using Larger models
>
> We experiment with MultiMix and Dense MultiMix using ViT-B/16 on Imagenet for 100 epochs using our experimental settings (Section 4.1). The baseline, MultiMix and Dense MultiMix achieves a top-1 accuracy of 78.12\%, 78.84\% and 79.53\% respectively. Thus, MultiMix and Dense MultiMix achieve a gain of 0.6\% and 1.4\% over the baseline. We shall add comparisons with SoTA mixup methods in the final paper.
>
> ### 5. Sensitivity to hyper-parameters
>
> By observing the plots of Fig. 4(a-d), we find that MultiMix and Dense MultiMix are not sensitive to hyperparameters. The plots are smooth and do not exhibit any local maxima on sharp peaks. Thus, only 4-5 measurements are enough to choose the best value for each hyperparameter. We have observed the same behavior in all our preliminary experiments while developing the method. This indicates the stability and robustness of our methods to the chosen configurations.

---

### Official Review · Reviewer_Xd45 · 2023-07-27

**Soundness:** 3 good
**Presentation:** 4 excellent
**Contribution:** 2 fair
**Rating:** 6
**Confidence:** 3

**Summary:**

The authors introduce MultiMix, an approach aimed at enhancing the Mixup class of techniques, which involve interpolating between input examples and their corresponding targets. This enhancement is achieved through the interpolation of a substantially more training examples, increased number of loss terms for each minibatch, and a more flexible interpolation factors. Further, the authors expand their proposed method with DenseMultiMix, a variation capable of interpolating examples in sequential data. Empirical evidence indicates that the proposed MultiMix and DenseMultiMix not only boost accuracy but also bolster the robustness metric across a diverse set of tasks, including but not limited to classification and object detection.

**Strengths:**

There's a noticeable diminishing return when interpolating additional input examples in Mixup. This presents a limitation to the advancement of Mixup-related methodologies, which focus on interpolating more training examples. However, the approach introduced in this paper appears to overcome this limitation by implementing interpolation in the embedding space, as opposed to the input space, evidenced by Fig 4c. I regard this as a significant contribution of this paper.

Furthermore, the adaptation of Mixup to sequential data like text has always been frustrated. There are three primary obstacles: 1. The blending of text tokens doesn't carry the same significance as mixed images. 2. How to interpolate along the temporal dimension or the spatial dimension. 3. Discrepancies in length among examples within a mini-batch. The introduced MultiMixup provides solutions to at least the first two challenges. Interpolating within the embedding space prevents the creation of nonsensical input tokens, while MultiMixup along with attention delivers an effective means of interpolating representations in the spatial dimension.

**Weaknesses:**

There are several weaknesses in this manualscript, as explaint in the following.

1. The motivation for the suggested method is insufficiently compelling within the manuscript. The only statement regarding to the motivation is ```In this work, we argue that a data augmentation process should increase the data seen by the model, or at least by its last few layers, as much as possible. ```

    This reasoning seems to be superficial as it fails to distinguish itself from the preceding data augmentation methods. Moreover, it doesn't elaborate on how the proposed approach enhances accuracy and robustness. More explicitly, while it's true that any data augmentation method would increase the volume of data processed by the model, not all such methods that do so effectively improve the model performance. Thus, the motivation fails to pinpoint the core of the research problem.

2. Regarding the empirical evaluation, it would be appropriate for the authors to include comparison with methodologies such as AugMix [1]. Similar to the proposed method, AugMix enhances the volume of data points processed by the model through interpolating examples derived from varying combinations of image pre-processors using the Dirichlet distribution. Moreover, it offers a substantial gain in both accuracy and robustness metrics. Also, the divergence loss could potentially counter the scarcity of dense labels.

     [1]: Hendrycks, Dan et al. “AugMix: A Simple Data Processing Method to Improve Robustness and Uncertainty.” ICLR 2020.

3. The performance improvement over the established baseline is rather modest. In such a scenario, it would be beneficial if the manuscript offered more profound insights to the community to better understand Mixup-oriented techniques. Such insights could include how the proposed method is connected to existing methods like AugMix, as well as an exploration of why the proposed methodology shows greater promise. For instance, while Section 4.4 and Figure 6 in the appendix provide useful information, they lack a comparison between the proposed method and existing methods, as well as a comparison among different variants of MultiMix as done in Section 4.5. Empirical insights of this would offer more compelling evidence that the proposed methodology is a valuable contribution to the community.

4. A less significant shortcoming pertains to the scope of the empirical evaluation. The batch size is held to be 128 in the domain of image classification. Understanding the robustness of the proposed method to batch size selection is crucial, especially as we move towards models with more # parameters where batch size tends to decrease. Specifically, when the batch size is small, MultiMix should offer more advantages over standard Mixup due to its ability to draw from a larger pool of interpolated examples. Another dimension worth exploring is the application of DenseMultiMix in the realm of text data. The manuscript would significantly benefit from showing evidence that the proposed method can address some of the challenges of extending Mixup-style techniques to the domain of text. I understand that interpolating dense text labels is a non-trivial task, preliminary studies on tasks such as text classification could suffice.

**Questions:**

Please see the weakness section.

**Limitations:**

One limitation as mentioned above is the application of DenseMultiMix in the realm of text data.

---

> ### Author Rebuttal · Authors · 2023-08-09
>
> We appreciate the valuable feedback from R-Xd45. We address the concerns as follows:
>
> ### 1. Motivation
>  Please refer to our comment on *Author Rebuttal by Authors*.
>
> ### 2. Comparison with AugMix
>
> Thanks for sharing this work.
>  **AugMix vs. MultiMix.** AugMix applies a convex combination of more than two examples, similar to MultiMix. The interpolation is between three distinct augmentations applied to an image, followed by a convex combination with the initial clean image. Similar to MultiMix, the interpolation weights are drawn from a Dirichlet distribution. However, AugMix interpolates examples in the *input space*, whereas MultiMix in the *embedding space*; the number of convex combinations is *limited to the batch size* in AugMix and *arbitrarily large* ($n=1000$ in practice) in MultiMix; and each convex combination is *limited to 3 examples* in AugMix and *as large as the mini-batch* in MultiMix.  Crucially, Fig. 4 shows that all 3 elements are needed to improve accuracy: embedding space, number $n$ of generated examples and number $m$ of examples being interpolated.  We shall add AugMix to the discussion of L55-61 and to Table 1. Then, AugMix will be closer to SuperMix [6].
>
>  **Experimental evaluation.** We follow the experimental setting of AugMix and evaluate the robustness of MultiMix and Dense MultiMix on CIFAR-100-C using WideResnet and ImageNet-C using Resnet-50. We report the classification error on CIFAR-100-C and mean corruption error on ImageNet-C. We also compare MultiMix and Dense MultiMix with AugMix using our experimental settings (Sec. 4.1) for image classification on CIFAR-100 using Resnet-18 and ImageNet using Resnet-50.  The classification accuracy results are summarized as:
> |Dataset|CIFAR100-C|ImageNet-C|CIFAR100|ImageNet|
> |-|-|-|-|-|
> |Input Mixup|50.4|72.8|79.8|77.4|
> |AugMix|35.9|68.4|80.1|77.7|
> |MultiMix(ours)| 33.8| 67.9| 81.8| 78.8|
> |Dense MultiMix(ours)| 31.7| 67.1| 81.9| 79.4|
>
> On robustness to corruptions, we observe that both MultiMix and Dense MultiMix outperform AugMix up to 4\% on CIFAR-100-C, and are also robust to corruptions on ImageNet-C. While AugMix is more robust to corruptions than Input Mixup, it does not bring a significant improvement in accuracy over CIFAR-100 and ImageNet. This limitation of AugMix is also validated in Table 6 (Appendix) of the AugMix paper. By contrast, MultiMix and Dense MultiMix improve classification accuracy and are robust to corruptions. We shall add these results to the final paper.
>
> ### 3. Additional empirical evidence
> We discuss the differences and compare MultiMix and SoTA mixup methods in Table 1, Sec. 2, Sec. 4.2 and 4.3, as well as Sec. A.2 and A.3. While the performance improvement over SoTA on classification is modest in Tables 2 and 3, it is significant on robustness in Table 4, reaching 3.5\%. We further show the following achievements of MultiMix and Dense MultiMix:
>
> - **Improving out-of-distribution detection and model calibration.** A standard benchmark for evaluating over-confidence of a model is to measure its ability to detect out-of-distribution examples. In Sec. A.4 and Table 9 of the Appendix, we compare MultiMix and Dense MultiMix against SoTA mixup methods for out-of-distribution detection. We show that although the gain of MultiMix and Dense MultiMix over SoTA mixup methods is small on image classification, they *significantly reduce over-confident incorrect predictions* and achieve superior performance on out-of-distribution detection, with gain ranging from 1.8 to 6.6\%.
>   Another way to evaluate over-confidence is *model calibration*. We evaluate calibration of the model using MultiMix and Dense MultiMix with Resnet-18 on CIFAR-100. We report *mean calibration error* (mCE) and *overconfidence error* (OE) below:
> |Metric|ECE|OE|
> |-|-|-|
> |Baseline|10.25|1.11|
> |Input Mixup|18.50 |1.42|
> |Manifold Mixup|18.41|0.79|
> |Co-Mixup|5.83|0.55|
> |AlignMixup|5.78|0.41|
> |MultiMix(ours)|5.63|0.39|
> |Dense MultiMix(ours)|5.28|0.27|
>
> Both MultiMix and Dense MultiMix achieve lower ECE and OE than SoTA methods, indicating that they lead to a better calibrated model.
>
> - **Generalizing to unseen target domains.** We show that MutliMix and Dense MultiMix generalizes to unseen target domains in our response to R-Hzfq -- *Novelty on Dirichlet's distribution and Comparison with DAML*.
>
> - **Improving inter-class separability.** We show that MutliMix and Dense MultiMix improve inter-class separability in our response to R-6vc1 -- *Metrics to improve class separability*.
>
> - **Dense classification loss improving SoTA methods.** In Sec. A.4 (Appendix) under ``Mixup methods with dense loss'', we study the effect of the dense loss on SoTA mixup methods. In Table 10, we show that on average, the dense loss brings a gain of 0.7\% over all mixup methods.
>
> ### 4. Robustness with smaller batch size
> Thanks for the suggestion. We compare Input Mixup, Manifold Mixup and MultiMix on CIFAR-100 using ResNet-18, using batch size $b<128$ for image classification. By default, we use $n=1000$ generated examples and $m=b$ examples being interpolated, following the experimental settings of Sec. 4.1. The results are as follows:
> |$m$ |2|25|50|100|
> |-|-|-|-|-|
> | Input Mixup|77.4| 78.3|79.0|79.5|
> | Manifold Mixup|78.6| 79.4|79.9|80.3|
> | MultiMix(ours)|80.9| 81.3|81.6|81.8|
>
> The accuracy increase of MultiMix with batch size $b$ is similar to the accuracy increase with number $m$ of examples being interpolated (Fig. 4(c)). This in expected as, in the new experiment, $m$ and $b$ are increasing together. We validate the reviewer's hypothesis that the performance improvement of MultiMix over Input or Manifold Mixup is higher for smaller batch size.
>
> ### 5. MultiMix for text classification
> Thanks for the suggestion. Dense MultiMix is not straightforward to apply to text, since text is represented by sequences of discrete tokens of variable length, which do not have a fixed spatial resolution. Such extension is interesting as future work.

---

> > ### Comment · Area_Chair_b7xB · 2023-08-14
> > **Check elements of the rebuttal**
> >
> > Dear Authors,
> >
> > Many thanks for your rebuttal.
> >
> > To make some progress in the discussion, could you please confirm the values of the table with CIFAR100-C	and ImageNet-C results?
> > Right now, they seem to indicate that `*MultiMix` would worsen the performance (which appears to be in contradiction with the text, if I am not mistaken)
> >
> > The AC of the paper.

---

> > > ### Author Response · Authors · 2023-08-15
> > > **Response to AC**
> > >
> > > Dear AC,
> > > Thank you for your initiative. Please note that ``lower is better`` for classification error (on CIFAR-100-C) and mean corruption error (on ImageNet-C), while ``higher is better`` for classification accuracy (on CIFAR-100 and Imagenet). Thus, our text is correct. We apologize if the table was not clear enough. We will make it clearer in the paper.

---

> > > > ### Comment · Area_Chair_b7xB · 2023-08-15
> > > >
> > > > @{Authors, Xd45}: Thanks for confirming.

---

> > ### Comment · Reviewer_Xd45 · 2023-08-18
> >
> > I have read other reviews and the author's rebuttal. I really appreciate the further explanation of the motivation and how MultiMix leads to better accuracy and robustness, and the additional experimental results. As a result, I will increase my rating to 'Weak Accept'.

---

> > > ### Author Response · Authors · 2023-08-19
> > > **Thank you R-Xd45**
> > >
> > > We sincerely appreciate R-Xd45's time and effort in evaluating our work, for the insightful comments, and for raising the score to "Weak Accept". We shall add the additional experiments and the motivation to the camera-ready version of the paper.

---

### Author Rebuttal · Authors · 2023-08-09

As summarized in our first contribution (L81-83), MultiMix consists of the following three elements:

- Increase the number $n$ of generated mixed examples beyond the mini-batch size $b$ (L46-54, L71-78, Table 1/column *terms*, Fig. 4(b)).
 - Increase the number $m$ of examples being interpolated from $m = 2$ (pairs) to $m = b$ (a single tuple containing the entire mini-batch) (L55-61, Table 1/column *mixed*, Fig. 4(c)).
 - Perform interpolation in the *embedding* space rather than the *input* space (L27-28, L43-45, Table 1/column *space*', Fig. 4(a)).

Dense Multimix (L66-78) is our second contribution (L84-85). It is an additional application of element 1, further increasing the number of generated mixed examples to $nr$ per mini-batch (L71-78, Table 1/column *terms*, Fig. 4(a-c)), where $r$ is the spatial resolution.

All three elements are important in achieving the performance of MultiMix beyond the SoTA. Removing any element leads to sub-optimal results, as shown in the ablation of Fig. 4(a-c). Previous works have missed this combination. For example, Manifold mixup [36] is element 3 alone (L27-28). A number of works [1,6,47] have attempted element 2 alone (L58-60) and found it not effective for $m > 3$. To our knowledge, element 1 has not been investigated before. As requested, we elaborate below on our motivation of element 1. More on the motivation of elements 2 and 3 follow in the responses to R-SS7F and R-6vc1, respectively.

`` R-Xd45: The only statement regarding to the motivation... fails to distinguish itself from the preceding data augmentation methods.``

Our motivation for element 1 is stated in L32-38, which we elaborate on here. The motivation stems from the original mixup paper [47], where the *expected risk* is defined as an integral over the underlying continuous data distribution. Since that distribution is unknown, the integral is approximated by a finite sum over the training data, i.e, the *empirical risk*, where each training example gives rise to one loss term. A better approximation is the *vicinal risk*, where a number of augmented examples is sampled from a distribution in the vicinity of each training example, thus *increasing the number of loss terms per training example*. Mixup applies the same idea to pairs of training examples, where the vicinal distribution of each training example is defined over linear segments to all other training examples. However, as a practical implementation, the authors of [47] *"convex combine a minibatch with a random permutation of its sample index.''* Thus, the opportunity of increasing the number of loss terms per training example is lost: each mini-batch of size $b$ gives rise to precisely $b$ loss terms. Unfortunately, this choice is adopted by all follow-up works on mixup.

In our work, *"we argue that a data augmentation process should increase the data seen by the model, or at least by its last few layers, as much as possible''* (L39-43). By this, we mean that even a small part of the model should see more inputs and targets than $b$ per mini-batch, giving rise to more loss terms than $b$ per mini-batch (L46-53). Our particular solution is that the classifier head $g_W$ sees $n \gg b$ mixed inputs and targets, giving rise to $n$ loss terms per mini-batch. This is clarified in column *terms* of Table 1, where MultiMix differs from all previous mixup works. The motivation is precisely that $n \gg b$ loss terms provide a better approximation of the expected risk integral. Even if the encoder $f_\theta$ only sees $b$ clean examples per mini-batch, it is still affected by backpropagating gradients from all $n$ mixed examples during training.

``R-Xd45: it doesn't elaborate on how the proposed approach enhances accuracy and robustness``

Using more loss terms than $b$ per mini-batch is not very common, at least not in classification. In *deep metric learning*, it is common to have a loss term for each pair of examples in a mini-batch, giving rise to $\frac{1}{2} b (b-1)$ loss terms per mini-batch $b$, even without mixup [1*]. In image classification, *supervised contrastive learning* [2*] involves a loss term for every positive pair of each example in the mini-batch, thus also resulting in more than $b$ loss terms per mini-batch. *Dense loss functions* (L116-122) also increase the number of loss terms per training example by a factor $r$, where $r$ is the spatial resolution.

We hypothesize that increasing the number of loss terms beyond $b$ per mini-batch improves the accuracy and robustness of the model by providing a better approximation of the expected risk integral. MultiMix is the first method to do so for mixup, increasing the number of loss terms per mini-batch to $n \gg b$. Dense MultiMix is the first dense mixup operation, further increasing the number of loss terms per mini-batch to $nr$ (Table 1/column ``terms''). Our hypothesis motivating element 1 is validated in Fig. 4(b), where accuracy increases continuously as we increase $n$, both for MultiMix and Dense MultiMix.

``R-6vc1 - it is hard to find a noticeable difference between the `trial' of the original Mixup paper and the proposed method except for the target space``

The `trial' of the original Mixup paper [47] refers to element 2, $m > 2$, where $m$ is the number of examples being interpolated. Interpolating in the embedding space refers to element 3. None of these two elements alone is the main contribution of our work. As stated above, a number of works [1,6,47] have attempted element 2 alone (L58-60), while Manifold mixup [36] is element 3 alone (L27-28). The main contribution is the observation that all three elements, where element 1 has not been attempted before. We have made our best effort to highlight element 1: L46-64, L71-78, Table 1/column *terms*.

[1*] Venkataramanan et al., It Takes Two To Tango: Mixup for Deep Metric Learning, ICLR 2022

[2*] Khosla et al., Supervised Contrastive Learning, NeurIPS 2020

---

### Author Response · Authors · 2023-08-14
**Request for Reviewers feedback**

Dear Reviewers,

We appreciate your time and effort in evaluating our work. If any concern remains unaddressed, we are happy to provide additional feedback.

Thank you,
Authors

---

### Decision · Program_Chairs · 2023-09-21

**Decision:**

Accept (poster)

**Comment:**

The reviewers and meta reviewer all carefully checked and discussed the rebuttal. They thank the authors for their response and their efforts during the rebuttal phase.

The reviewers and meta reviewer all acknowledge the original and sound contribution of combining different (subtle) aspects of mixup that had been considered so far in isolation and unsuccessfully. Similarly, the reviewers and meta reviewer all call out the overall good quality of the manuscript (writing, structure).

The rebuttal has considerably strengthened some aspects of the work (e.g., comparison with AugMix and DAML, analysis of smaller batch sizes, sensitivity with respect to the selection of the hyperparameters). However, there are still some remaining important concerns. As a result, the reviewers and the meta reviewer are weakly inclined to accept the paper.

In particular, the authors are urged to carefully update their final manuscript with the following points:

* _“Third, instead of using a single scalar value of lambda per mini-batch….”_: Using different lambda’s within a single batch seems to have been used by default in previous work with standard mixup, e.g., [A]
* _“...Using more loss terms than per mini-batch is not very common…”_: It should be mentioned that in the literature on efficient ensembles (e.g., [B, C, D, E]), the batch size with b examples is often turn to M*b loss terms, where M denotes the number of ensemble members.
* Systematically add the additional empirical evidence, and corresponding discussions, obtained during the rebuttal phase
* _“...MultiMix is thus a better approximation of the expected risk integral…”_: This is an argument that has been repeatedly used during the rebuttal. It is thus important to provide more insights and an empirical analysis (e.g., based on a synthetic setting where the true risk integral can be empirically estimated) and/or some theoretical arguments.
* The discussion with reviewer R-SS7F suggests that the hyperparameters have been selected in a *test-aware* fashion (on CIFAR-100, before then being identically applied to other settings). This departs from a clean 3-fold (train, validation, test) selection. As a result, the analysis started in https://openreview.net/forum?id=HKueO74ZTB&noteId=F5gh77Hr2x should be extended to all other hyperparameters.
* What happens with larger batch sizes? A study with b > or >> 128 (as used in many standard training procedures) would help understand if the benefits of the proposed approach plateau

If the paper was submitted to a journal, it would be accepted conditioned on those key changes, the meta reviewer thus expects all those changes to be carefully implemented.

[A] Carratino et al., 2021, On Mixup Regularization. (https://github.com/google/uncertainty-baselines/blob/main/baselines/imagenet/deterministic.py#L64)

[B] Wen et al, 2019, BatchEnsemble: An Alternative Approach to Efficient Ensemble and Lifelong Learning

[C] Dusenberry et al, 2020, Efficient and Scalable Bayesian Neural Nets with Rank-1 Factors

[D] Havasi et al, 2021, Training independent subnetworks for robust prediction

[E] Allingham et al, 2021, Sparse MoEs meet Efficient Ensembles